**Effect of plateau pikas disturbance and patchiness on ecosystem carbon emission of**
**alpine meadow on the northeastern part of Qinghai-Tibetan Plateau**
Yu Qin[1], Shuhua Yi[2, 1*], Yongjian Ding[1, 3], Wei Zhang[1, 3], Yan Qin[1, 3],Jianjun Chen[4,5], Zhiwei
Wang[1,6]
1. State Key Laboratory of Cryospheric Sciences, Northwest Institute of Eco-Environment
and Resources, Chinese Academy of Sciences, 320 Donggang West Road, Lanzhou 730000,
China
2. School of Geographic Sciences, Nantong University, 999 Tongjing Road, Nantong, Jiangsu,
226007, China
3. University of the Chinese Academy of Sciences, No.19A Yuquan Road, Beijing 100049,
China
4.College of Geomatics and Geoinformation, Guilin University of Technology, 12 Jiangan Ro
ad, Guilin, 541004, China
5. Guangxi Key Laboratory of Spatial Information and Geomatics, 12 Jiangan Road, Guilin, 5
41004, China
6. Guizhou Institute of Prataculture, Guizhou Academy of Agricultural Sciences, Guiyang,
550006, People's Republic of China
* E-mail: yis@lzb.ac.cn
Tel: +86-931-4967356

**Abstract**

Plateau pikas (*Ochotona curzoniae*) disturbance and patchiness intensify the spatial heterogeneous distribution of vegetation productivity and soil physicochemical properties, which may alter ecosystem carbon emission process. Nevertheless, previous researches have mostly focused on the homogeneous vegetation patches rather than heterogeneous land surface. Thus, this study aims to improve our understanding of the difference in ecosystem respiration (Re) over heterogeneous land surface in an alpine meadow grassland. Six different land surface: large bald patch, medium bald patch, small bald patch, intact grassland, above pika tunnel and pika pile were selected to analyze the response of Re to pikas disturbance and patchiness, and the key controlling factors. The results showed that (1) Re under intact grassland were 0.22-1.07 times higher than pika pile and bald patches; (2) soil moisture (SM) of intact grassland was 2-11% higher than those of pika pile and bald patches despite pikas disturbance increased water infiltration rate, while soil temperature (ST) under intact grassland was 1-3℃ less than pika pile and bald patches; (3) Soil organic carbon (SOC) and total nitrogen (TN) under intact grassland were approximate 50 % and 60 % less than above pika tunnel, whereas 10-30 % and 22-110 % higher than pika pile and bald patched; and (4) Re was significantly correlated with SM, TN and vegetation biomass (P<0.05). Our results suggested that pikas disturbance and patchiness altered ecosystem carbon emission pattern, which was mainly attributed to the reduction of soil water and supply of substrates. Given that the wide distribution of pikas and large area of bald patches, the varied Re under heterogeneous land surfaces should not be neglected for estimation of ecosystem carbon emission at plot or region scale.

**Keywords**: pikas disturbance; patchiness; ecosystem respiration; alpine meadow; the Qinghai-Tibetan Plateau

**Introduction**

Ecosystem respiration (Re) is the key process to determine the carbon budget in the terrestrial ecosystem. Thus, even a small imbalances between $CO_2$ uptake via photosynthesis and $CO_2$ release by ecosystem respiration can lead to significant interannual variation in atmospheric $CO_2$ (Schimel et al., 2001; Cox et al., 2000; Grogan and Jonasson, 2005; Oberbauer et al., 2007; Warren and Taranto, 2011). Dependent on autotrophic (plant) and heterotrophic (microbe) activity, ecosystem respiration is mainly controlled by abiotic factors (primarily temperature and water availability) (Chimner and Welker, 2005; Flanagan and Johnson, 2005; Nakano et al., 2008; Buttlar et al., 2018), and supply of carbohydrate fixed by leaves, vegetation litter and soil organic matter (Janssens et al., 2001; Reichstein et al., 2002). Therefore, any external disturbance altering environmental conditions and affecting vegetation growth would exert profound influence on ecosystem carbon emission.

One of the basic function of terrestrial ecosystem is to regulate carbon balance between the atmosphere and ecosystem (Canadell et al., 2007; Le Quéré et al., 2014; Ahlström et al., 2015). However, this balance would be broken by widespread land degradation (Post and Kwon, 2000; Dregne, 2002), which accompanied with the reduction of photosynthetic fixed carbon dioxide from atmosphere and carbon sequestration by soils (Defries et al., 1999; Upadhyay et al., 2005). It was estimated that land degradation had resulted in 19-29 Pg C loss worldwide (Lal, 2001). Over the past decades, grasslands have experienced patchiness throughout the world and this process is still ongoing (Baldi et al., 2006; Wang et al., 2009; Roch and Jaeger, 2014). Patchiness generally refers to a landscape that consists of remnant areas of native vegetation surrounded by a more heterogeneous and patchy situation (Kouki and Löfman, 1998). Other than climate change (Yi et al., 2014), vegetation self-organization (Rietkerk et al., 2004; Venegas et al., 2005; McKey et al., 2010) or anthropogenic disturbances (Kouki and Löfman, 1998; Yi et al., 2016), rodents burrowing activities were also considered as the origin of the patchiness (Wei et al., 2006; Davidson and Lightfoot, 2008). This patchiness intensified spatial heterogeneity of land surface and led to the changing of the structure and function of the original ecosystem (Herkert et al., 2003; Bestelmeyer et al., 2006; Lindenmayer and Fischer, 2013). For instance, there is abundant evidence that patchiness not only intensified the spatial heterogeneous distribution of

ecosystem organic carbon (C) and vegetation productivity (Yan et al., 2016; Qin et al., 2018)
but also altered the pattern of coupled water and heat cycling between the land surface and the
atmosphere (Saunders et al., 1991; You et al., 2017; Ma et al., 2018). Consequently, this may
alter ecosystem carbon emission process (Juszczak et al., 2013).
Plateau pikas (*Ochotona curzoniae*, hereafter pikas) are small mammals endemic to the
alpine grasslands on the Qinghai-Tibetan Plateau (QTP) (Smith and Foggin, 1999; Lai and
Smith, 2003). Living in underground, they excavated deep layer soil to surface through
foraging and digging activities (Lai and Smith, 2003) and led to substantial bald piles on the
ground. The bald pile was considered to gradually become bald patches under soil erosion,
gravity, freeze-thaw and other factors (Chen et al., 2017; Ma et al., 2018). As a consequence,
natural vegetation patches and adjacent bald patches with different sizes, and pikas piles
represent the most common landscape pattern in the alpine meadow grassland on the QTP.
Previous studies have demonstrated that pikas disturbance and patchiness weaken the function
of alpine meadow as a carbon sink (Liu et al., 13; Peng et al., 2015; Qin et al., 2018) and
accelerated ecosystem carbon emission rate (Qin et al., 2015a). Nevertheless, most of these
studies have mainly focused on ecosystem carbon emission rate under the homogeneous land
surface rather than heterogeneous land surfaces. Thus, the specific aims of this study were to
(1) investigate the spatial heterogeneity of Re among different surface types (plateau pika pile,
above pika tunnel, different sizes of bald patches and vegetation) of alpine grassland; (2)
illuminate the potential regulating mechanism of pikas disturbance and patchiness to
ecosystem respiration (Re) in an alpine meadow grassland in the northeastern part of
Qinghai-Tibetan Plateau (QTP).
**Materials and methods**
**Site description**
This study was conducted at the permanent plots at Suli Alpine Meadow Ecosystem
Observation and Experiment Station (98°18'33.2", 38°25"13.5', 3887 m a.s.l.), Northwest
Institute of Eco-Environment and Resources, Chinese Academy of Science. The study area is
characterized by a continental arid desert climate, with low mean annual air temperature, little
rainfall, and high evaporation (Wu et al., 2015). The mean annual air temperature was
approximately -4°C and the annual precipitation ranged from 200 to 400mm, respectively
(Chang et al., 2016). The permafrost type at our site is transition and the active layer depth is
$2.78 \pm 1.03$ m (Chen et al., 2012). The dominant plant species in the study area were *Kobresia*
*capillifolia*, *Carex moorcroftii* (Qin et al., 2014). Soils was classified as "felty" with a pH of
8.56, 30.96 % silt and fine, 57.52 % fine sand and 10.68 % coarse sand, and soil bulk density
is 1.41 g cm$^{-3}$ within a 0-40 cm depth of the soil layer (Qin et al., 2015b). The grassland in
this area suffered from degradation due to permafrost degradation and external disturbance
from grazing livestock and small mammals, i.e. plateau pikas (Yi et al., 2011, Qin et al.,
2015a). As a result, a mosaic pattern of vegetation patches, bald patches with different sizes
and pika piles was common.

**Field observation**

At early June 2016, three 100 m × 100 m plots were established as replicates. Each 100 × 100
m plot was in a distance of less than 50 m, which has the similar plant and terrain. In each
plot, six representative land surfaces were selected: (1) large bald patch with size larger than
9.0 m$^2$ (LP), (2) medium bald patch with size of 1.0-9.0 m$^2$ (MP), (3) small bald patch with
size of less than 1.0 m$^2$ (SP), (4) intact grassland patch (IG), (5) above pika tunnel (PT), (6)
old pika pile (PP) (Figure 1) (Yi et al., 2016; Qin et al., 2018). There were no other mammals,
e.g. marmot and zokor in our study plots. All of the piles in each plot were created by plateau
pikas. They were distinguished easily in aerial photographs. Large bald patches had less
vegetation cover and the smallest side was larger than 3 m. Medium patches also covered by
less vegetation cover and the largest side was in a range of 1 to 3 m and small bald patches
were characterized by less vegetation cover and the largest side was less than 1 m. Intact
grassland was characterized by high vegetation cover and no large and medium bare land was
found. Pika tunnel and pika pile usually co-existed. Pika tunnel is approximately 6 cm in
diameter and pika pile is in the front of pika tunnel, 60 cm in diameter and less vegetation
cover. We calculated the threshold area of large, medium and small patches by aerial
photograph. Each aerial photograph has 12 million pixels. At a height of 20 m, the resolution
of each pixel is ~1 cm and each photograph covers ~26 m × 35 m of ground. Pixels in each
aerial image were first classified into two groups, i.e. vegetated or bare patches (Yi, 2017).
Then patches with different sizes were created using OpenCv Library. And finally, fractions
of vegetation and bare patches (large, medium and small patches) were calculated. For each
surface type in each plot, six 1 m × 1 m quadrats were set up, of which three was used for soil
saturated hydraulic conductivity measurement and three for soil compactness measurement,
soil and vegetation sampling. We also set up another three 1 m × 1 m quadrats and three 2 m
× 2 m quadrats in each surface type in a 100 m × 100 m plot for measuring soil temperature,
soil moisture and ecosystem respiration.

(Insert Figure 1 here)

A meteorological tower was established in our observation station since 2008. Air

temperature (°C) at 2.0m was measured by HMP45C (Vaisala, Helsinki, Finland), and
precipitation was measured using an all-weather precipitation gauge (Geonor T-200B,
Norway) (Wu et al., 2015). Soil temperature and moisture at 10 cm were measured by using
an auto-measurement system (Decagon Inc., USA) from early June to the late August. The
system consisted of an EM50 logger and five 5TM sensors. The data were logged
automatically every 30 minutes. Soil saturated hydraulic conductivity and compactness were
measured one time in each month from June to August. Soil saturated hydraulic conductivity
was measured by Dual Head infiltrometer (Decagon Inc., USA). The measurement process
included soak time 15 minutes, hold time 20 minutes at low pressure head (5 cm) and high
pressure head (15 cm) with 2 cycles. Each measurement takes 95 minutes altogether. Soil
compactness was measured with TJSD-750 (Hangzhou Top Instrument co., LTD, Hangzhou,
China) from the soil surface to 10 cm depth. Ecosystem respiration rates were measured using
the LICOR-8150 Automated Soil $CO_2$ Flux System, which was an accessory for the
LI-8100A could connect 16 individual chambers at one time and were sampled and controlled
by the LI-8100A Analyzer Control Unit. The air temperature inside of the chamber was
measured using the internal thermistor of the chamber. The ecosystem $CO_2$ fluxes were
calculated by the equation as follow.
$$Fc = \frac{10VP_0\left(1-\dfrac{W_0}{1000}\right)}{RS(T_0+273.15)}\frac{\partial C'}{\partial t}$$

where $Fc$ is the soil $CO_2$ efflux rate (µmol m$^{-2}$ s$^{-1}$), $V$ is volume (cm$^3$), $P_0$ is the initial pressure
(kPa), $W_0$ is the initial water vapor mole fraction (mmol mol$^{-1}$), $R$ is the ideal gas constant (J
K$^{-1}$mol$^{-1}$), $S$ is soil surface area (cm$^2$), $T_0$ is initial air temperature (°C), and $\partial C'/\partial t$ is the initial
rate of change in water-corrected $CO_2$ mole fraction ($\mu mol^{-1}$ $s^{-1}$).
Six LICOR-8100-104 long-term opaque chambers (20cm in diameter LICOR, Inc.,
Lincoln, NE, USA) were used to measure alternately between three replicates for six land
surface types. Therefore, 3 days at least were required to complete one rotation measurements
of ecosystem respiration. To measure ecosystem respiration, eighteen polyvinyl chloride
collars with a 20 cm inner diameter and a 12 cm height were inserted into the soil with 3-4 cm
exposed to the air (Qin et al., 2013). All of the collars were installed at least 24 h before the
first measurement to reduce disturbance-induced ecosystem $CO_2$ effluxes. Ecosystem
respiration rates were measured every 7-10 days from June 16 to August 20 in 2016
depending on weather conditions. A round-the-clock measurement protocol was carried out
and ecosystem respiration rates were measured every 30 minutes. Each measurement takes 1
minute and 45 seconds, including pre-purge 10 seconds, dead band 15 seconds, observation
length 1 minute and post-purge 20 seconds.
**Soil and vegetation sampling**
Soil samples were collected during the periods of late July to early August 2016. In each
surface type of each plot, five soil cores were collected using a stainless-steel auger (5 cm in
diameter) at depths of 0-10, 10-20, 20-30 and 30-40 cm, and bulked as one composite sample
for each depth in each quadrat. Another five soil cores were sampled by cylindrical cutting
ring (7 cm in diameter and 5.2 cm in depth) to determine soil bulk density from each land
surface type. Pika tunnel was approximate 6 cm in diameter and 40 cm in depth. Therefore,
soil samples were available to collect at depth of 40cm. Totally, 512 soil samples were
collected. Soil samples were firstly air-dried, then removed gravel and stone with manual
sieving and finally weighed. The remaining soil samples with diameter less than 2 mm were
ground to pass through a 0.25 mm sieve for analysis of soil organic carbon (SOC) and soil
total nitrogen (TN) concentration. SOC was measured by dichromate oxidation using
Walkley-Black acid digestion (Nelson and Sommers, 1982). TN was determined by digestion
and then tested using a flow injection analysis system (FIAstar 5000, Foss Inc., Sweden).
Aboveground and belowground biomasses were determined within a 1 m × 1 m quadrat on 4
August 2016 during peak biomass and species diversity. There were a total of 108
aboveground and belowground vegetation samples (3 plots × 6 land surface types × 3
replicates) from the study area. Aboveground biomass was determined by clipping all
above-ground living plants at ground level, drying (oven-dried at 65°C for 48 h) and weighing.
Belowground biomass was sampled by collecting five soil columns, and each soil column was
5 cm in diameter and 40 cm in depth. Soil cores were washed with a gentle spray of water
over a fine mesh screen until soil separated from the roots, and then drying (oven-dried at
65°C for 48 h) and weighing.
**Statistical analysis**
The soil organic C and total N densities in different land surface were calculated using the
equation (2) and (3):
$$SOC = \sum_{i=1}^{n} \rho * (1 - \sigma_{gravel}) * C_{SOC} * D_i \qquad (2)$$

$$TN = \sum_{i=1}^{n} \rho * (1 - \sigma_{gravel}) * C_{TN} * D_i \qquad (3)$$

where SOC is soil organic C density (kg m$^{-2}$), TN is soil total N density (kg m$^{-2}$), $\rho$ is the soil
bulk density (g cm$^{-3}$), $\sigma_{garvel}$ is the relative volume of gravel (% w/w), $C_{SOC}$ is soil organic C
content (g kg$^{-1}$), $C_{TN}$ is soil total N content (g kg$^{-1}$) and $D_i$ is soil thickness (cm) at layer i,
respectively; i=1, 2, 3 and 4.
The data were presented as mean ± standard deviation. Statistical analyses were performed
using the SPSS 17.0 statistical software package (SPSS Inc., Chicago, IL, USA). One-way
analysis of variance (ANOVA) and a multi-comparison of a least significant difference (LSD)
test were used to determine differences at the p=0.05 level. The relationships of ecosystem
respiration with biotic and abiotic factors were analyzed by Pearson correlation analysis using
R.
**Results**
**Ecosystem respiration**
Ecosystem respiration showed significant difference among varied land surface types during
the growing season (Table 1, P<0.001). Except for the pika pile, ecosystem respiration
maximized in August and minimized in June (Figure 2). In June, ecosystem respiration under
intact grassland, above pika tunnel, small patch and pika pile had no significant difference and
the lowest ecosystem respiration was found under large and medium patches (Figure 2).
Average ecosystem respiration under intact grassland was 4.01 $\mu mol\ m^{-2}\ s^{-1}$ in July, which
was 24.35 % to 137.39 % higher than other surface types (Figure 2). In August, average
ecosystem respiration were 4.07 $\mu mol\ m^{-2}\ s^{-1}$ and 4.85 $\mu mol\ m^{-2}\ s^{-1}$ for intact grassland and
above pika tunnel, 2.59-3.81 $\mu mol\ m^{-2}\ s^{-1}$ for bald patches and 1.18 $\mu mol\ m^{-2}\ s^{-1}$ for pika pile
(Figure 2).
(Insert Table 1, Figure 2 here)
**Microclimate and soil hydrothermal characteristics**
Mean temperature and total rainfall during the growing seasons from 1 May to 30 September
in 2016 were 6.18 °C and 343.4 mm, respectively (Figure 3). Soil temperature and moisture
were significantly different among various land surface types (Table 1, P<0.05). The monthly
average soil temperature was in a range of 8.20-13.72 °C during June to August, which was
approximate 1-3 °C higher under pika pile and bald patches than the intact grassland (Figure
4a, P<0.05). The monthly mean soil moisture from June to August was approximate 30 % for
intact grassland and above pika tunnel, 25 % for small patch and pika pile, and 20 % for
larger and medium patch (Figure 4b). Soil saturated hydraulic conductivity also showed
significant variation under different land surface types (P=0.027, Table 2). For example, soil
saturated hydraulic conductivity under large bald patch, medium bald patch, small bald patch,
intact grassland patch, above pika tunnel and old pika pile were 1.54, 1.53, 2.14, 2.13, 2.12
and 2.58 cm $h^{-1}$, respectively (Figure 5). Soil saturated hydraulic conductivity under intact
grassland patch was approximate 40 % higher than medium and large patches and 17 % lower
than pika pile, while it was no significant difference among intact grassland patch, small patch
and above pika tunnel (P>0.05).
(Insert Table 2, Figure 3 to 5 here)
**Soil and vegetation properties**
Soil and vegetation properties showed significant variation under different land surface types
(Table 2) (P<0.001). Soil compactness was over 0.30 Pa in intact grassland and above pika
tunnel, approximate 0.20 Pa for bald patches and less than 0.10 Pa for pika pile (Figure 6),
respectively. Mean SOC and TN density under intact grassland were 52.45 % and 59.14 %
less than above pika tunnel, whereas they were 9.69-30.12 % and 22.47-109.62 % higher than
pika pile and bald patches (Figure 7). Aboveground and belowground biomass under intact
grassland were approximate 30 % higher than above pika tunnel, 90 % higher than pika pile,
123-252 % and 134-289 % higher than bald patches (Figure 8a, b).

(Insert Figure 6 to 8 here)

**Factors regulate ecosystem respiration**
We analyzed the relationships of ecosystem respiration with biotic and abiotic factors for six
land surface types (Figure 9). Correlation analysis showed that ecosystem respiration had no
significant correlation with soil temperature ($P>0.05$, Figure 9). However, ecosystem
respiration was significantly and positively related to soil moisture ($P<0.01$), soil total
nitrogen ($P<0.05$), aboveground ($P<0.05$) and belowground biomass ($P<0.05$) (Figure 9).

(Insert Figure 9 here)

**Discussion**
**Effect of pikas disturbance on ecosystem respiration**
Pikas burrowing activities increased oxygen content in deep soil, which contributed to the
decomposition of soil organic matter (Martin, 2003). The deposition of urine and feces by
small herbivorous mammals could also promote ecosystem nutrition circulation (Clark et al.,
2005). It was suggested that excreta deposited by pikas and frequently haunted in or near their
burrows supplied organic C available to microbial decomposition with an increase in
ecosystem $CO_2$ emission (Cao et al., 2004). Indeed, SOC and TN densities reached up to
14.54 and 0.98 kg m$^{-2}$ in above pika tunnel, which was 2.45 and 2.10 times higher than that of
intact grassland (Figure 7), respectively. The consistent results reported that the contents of
available soil nutrients around the pikas burrow were higher than those in control sites on an
alpine meadow (Zhang et al., 2016). We also found that SOC and TN densities under pika pile
decreased 13.35 % and 42.93 % than intact grassland. This was because pika burrowing
activity transferred of deeper, nutrient-poor soil to the soil surface, improved soil aeration
increased rate of organic carbon mineralization and soil erosion took away soil nutrition (Wei
et al., 2006; Qin et al., 2015a; Chen et al., 2017). However, except July, no significant
difference of Re was found between intact grassland and above pika tunnel, while Re under
pika pile was 42.08 % less than intact grassland (Figure 2). The similar result was also found
in an alpine meadow on the QTP (Peng et al., 2015), which indicated that ecosystem
respiration decreased with increasing of pika holes because of grassland biomass regulated
soil C and N with increasing number of pika holes. These results confirmed that pikas
disturbance did not increase ecosystem carbon emission directly, but facilitated $CO_2$ emission
into the atmosphere through pika holes (Qin et al., 2015a). The difference of ecosystem
respiration between intact grassland and pika piles was mainly related to changes in
vegetation biomass and soil moisture. For example, both aboveground and belowground
biomass decreased 244.62 % and 279.89 % under pika piles compared with the intact
grassland (Figure 8). The reduction of vegetation biomass production decreased aboveground
plant respiration and root respiration by decreasing carbon allocation (e.g., root exudates and
litter, and available SOC) (Raich and Potter, 1995; Högberg et al., 2002; Yang et al., 2018).
Consistent with previous studies which demonstrated that pikas burrowing activity increased
water infiltration rate (Hogan, 2010; Wilson and Smith, 2015), our results also showed that
soil saturated hydraulic conductivity in pika pile was significantly higher than bald and
vegetation patches (Figure 5). Nevertheless, the increased water infiltration was unable to
increase soil moisture under pika piles. For example, soil moisture under pika piles was
approximate 5 % lower than intact grassland (Figure 4). Our result was discrepant with
previous studies which reported old pika mound had the highest soil moisture during the
summer (Ma et al., 2018) and moderate pika burrowing activities increased surface soil
moisture (Li and Zhang, 2006). This difference may be contributed to the high pika density in
alpine meadow (Guo et al, 2017). Moreover, pika piles were loose (Figure 6) with less
vegetation cover (Figure 8), which was not beneficial for soil moisture storage.
**Effect of patchiness on ecosystem respiration**
Our results clearly showed that patchiness resulted in significant reduction of ecosystem
carbon emission. Compared with the intact grassland, ecosystem respiration decreased
approximate 17-48 % for bald patches (Figure 2). Two possible mechanisms could account
for the effects of patchiness on ecosystem respiration. On one hand, the reduction of SOC and
TN decreased microbial respiration by decreasing substrate supply to microbes in the
rhizosphere (Nobili et al., 2001; Scott-Denton et al., 2010). Our results indicated that
patchiness caused evident loss of SOC and TN (Figure 7) due to reduction in C input from
vegetation and increasing in C output from soil erosion (Qin et al., 2018). Previous study have
shown that the spatial heterogeneity of soil respiration was attributed to uneven soil organic
carbon and total nitrogen content (Xu and Qi, 2010). Soil organic carbon was considered as
the basic substrate of $CO_2$ emission by microbial decomposition (Sikora and Mccoy, 1990)
and soil total N enhanced ecosystem $CO_2$ emission by providing a source of protein for
microbial growth (Tewary et al., 1982). On the other hand, low moisture availability would
limit microbial respiration by restricting access to C substrates, reducing the diffusion of C
substrates and extracellular enzymes, and limiting microbial mobility (Yuste et al., 2003;
Wang et al., 2014). Our results showed that soil moisture under large and medium patches
decreased 10 % than intact grassland (Figure 4). Previous studies had reported that the soil
compaction of bald patches decreased the rate of water infiltration (Wuest et al., 2006; Wilson
and Smith, 2015), which was similar with our results showed that bald patches had less
saturated soil hydraulic conductivity (Figure 5). Low vegetation cover under bald patches was
not beneficial for water retention and utilization, where most of soil water was mainly lost as
a way of evaporation (Yi et al., 2014). We have measured evaporation of the intact grassland,
isolate grassland, large patches, medium patches and small patches since the early June 2016.
Three years results indicated that evaporation under bald patches were higher than the intact
grassland (data were not shown here).
**Factors affected ecosystem respiration**
Most previous studies showed that soil temperature explained most of the temporal variation
of ecosystem respiration on the alpine grassland on the QTP (Lin et al, 2011; Qin et al., 2015c;
Zhang et al., 2017). Our results indicated that soil temperature under pika piles and bald
patches was approximate 1 to 3 °C higher than intact grassland (Figure 4), which mainly
resulted from the heterogeneity of surface albedo, surface soil water retention, heat
conduction properties and radiation (Beringer et al., 2005; Pielke, 2005; Yi et al., 2013; You et
al., 2017). It was suggested that pikas disturbance create a better soil temperature buffer for
them to avoid the extreme cold in winter (Ma et al., 2018), whereas high soil temperature
under bald patch was a disadvantage for the recovery of vegetation because patch surface had
the smallest soil moisture content (Figure 4) and the largest daily range of soil temperature
(Ma et al., 2018). It was well known that rising of soil temperature under natural condition
enhanced ecosystem respiration by stimulating decomposition of soil organic matter (Conant
et al., 2008), increasing plant biomass (Yi et al., 2014) and activity of microbial enzymes
(Bond-Lamberty and Thomson, 2010). However, obvious relationship between Re and soil
temperature was not found in the present study (Figure 9), which suggested that other factors
involved in controlling Re induced by pikas disturbance and patchiness. Our results showed
that Re were positively correlated with soil moisture, soil total nitrogen, aboveground and
belowground biomass (Figure 9). Pikas disturbance and patchiness led to the drying and
loosening of soil (Figure 4 and 6). It was considered that loose, dry surface sediments and
strong winds were the primary factors responsible for soil erosion (Dong et al., 2010b) and
wind erosion was especially common in arid and semi-arid regions (Zhang and Dong, 2014).
This resulted in the reduction of soil organic carbon, total nitrogen and vegetation biomass
(Figure 7 and 8). The alteration of these biotic and abiotic factors induced by pikas
disturbance and patchiness led to the decline of ecosystem respiration. Nevertheless, the
decline of ecosystem respiration did not completely offset the sequestration of C fixed by
photosynthesis because of the lower vegetation cover under bald patches and pika piles.
Given the large area covered by bald patches in alpine grasslands, patchiness was more
susceptible to erosion and exert greater influence on ecosystem respiration than pikas
disturbance. Recent study has also reported that bald patches of various sizes on the
grasslands played a much more important role than pikas direct disturbance in reducing
vegetation cover, aboveground biomass, soil carbon and nitrogen (Yi et al., 2016).
**Effect of pikas disturbance on patchiness**
Natural vegetation patches, bald patches with different sizes and pikas piles coexisted on the
alpine meadow (Figure 1), which supported that alpine grassland had also experienced
fragmentation (Qin et al., 2018). Several proposed mechanisms may be accounted for the
formation and development of patchiness in alpine grassland. As one of dominant form of
land utilization, alpine grasslands are widely used for grazing. Previous studies suggested that
overgrazing destroyed the original vegetation and led to decrease in the coverage and
looseness of soil (Dong et al., 2013), which was prone to form bald patch due to soil erosion
(Fécan et al., 1998; Zhang and Dong, 2014). Other than livestock, alpine grassland is also
habitats for many small mammals such as plateau pika, zokor (*Eospalax fontanierii*), marmot
(*Marmota himalayana*) and fox (*Vulpes ferrilata*). Pikas were considered to create a patchy
matrix by changing soil properties (Chen et al., 2017), digging tunnels and burying activities
(Dong et al., 2013). On one hand, pikas bury vegetation by fresh excavated soil, then small
bare soil patches are formed and further large soil patches are then formed by linking small
bare soil patches by wind and/or water (Wei et al., 2007; Ma et al., 2018). On the other hand,
pikas dig tunnel underground. Although pikas make burrows are the primary homes to a wide
variety of small birds and lizards (Smith and Foggin, 1999), the collapse of pika tunnels
results in the emergence of bald soil patches (Zhou et al., 2003; Cao et al., 2010). Moreover,
alpine grassland is underlain by extensive permafrost (Chen and Wu, 2007). The repeated
freeze and thaw cause the crack of the sod around the barren area (Yang et al. 2003) and
create precondition for forming bald patch. However, to date, there are no direct evidences to
demonstrate the potential mechanism for forming and developing of patchiness for alpine
grassland on the QTP. It is, therefore, critical to perform long-term repeated monitoring
studies to determine whether bald patches are developed from pika piles or burrow tunnels
and what the major factors affecting bald patch expansion are (Yi et al., 2016).
**Conclusions**
In this study, we investigated soil physicochemical properties, vegetation biomass and
ecosystem respiration (Re) under six land surfaces originating from pikas disturbance and
patchiness. We also analyzed the dominant factors regulated the Re. Our results showed that
pikas disturbance and patchiness decreased soil moisture but increased soil temperature,
which may be conducive to pikas survive in cold season but disadvantage for vegetation
growth. Patchiness caused evident decreasing in SOC and TN density, while both SOC and
TN density showed different response under pika piles and burrows. Both pikas disturbance
and patchiness decreased ecosystem carbon emission, and ecosystem respiration sharply
correlated with soil moisture, TN and vegetation biomass. Our results indicated that pikas
disturbance and patchiness led to the changing of ecosystem respiration process owing to the
drying of soil and the reduction of substrate supply. However, the decline of ecosystem
respiration may not able to offset the sequestration of C fixed by photosynthesis.
**Acknowledgment**
The authors would like to thank Mr. Jun Zhang and Bingbing Bai for their help in field
sampling. This study was jointly supported by grants from the National Key R&D Program of
China (2017YFA0604801), the National Natural Science Foundation of China (41501081 and
41690142), the independent grants from the State Key Laboratory of Cryosphere Sciences
(SKLCS-ZZ-2018) and science and technology support program of Science and Technology
Agency in Guizhou "The Key technology and engineering demonstration of farmland system
control and restoration in Tongren mercury polluted area" (Qiankehezhicheng[2017]2967).

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

abstract).

**Table 1.** ANOVA results of soil temperature, soil moisture and ecosystem respiration under
different land surface types.

| | Soil temperature | | | Soil moisture | | | Ecosystem respiration | | |
|---|---|---|---|---|---|---|---|---|---|
| | June | July | August | June | July | August | June | July | August |
| $F$ | 8.614 | 10.955 | 1.806 | 387.472 | 210.878 | 97.060 | 5.270 | 10.447 | 8.855 |
| $P$ | <0.001 | <0.001 | 0.106 | <0.001 | <0.001 | <0.001 | 0.001 | <0.001 | <0.001 |

**Table 2.** ANOVA results of soil compactness, aboveground biomass, belowground biomass,
soil hydraulic conductivity, SOC and TN density under different land surface types.

| | Soil compactness | Aboveground biomass | Belowground biomass | Saturated hydraulic conductivity | SOC density | TN density |
|---|---|---|---|---|---|---|
| $F$ | 81.506 | 6.193 | 12.925 | 2.752 | 145.942 | 50.567 |
| $P$ | <0.001 | 0.002 | <0.001 | 0.027 | <0.001 | <0.001 |


**Figure legends**
**Figure 1.** An aerial photo of field observation of ecosystem respiration at six surface types: (1)
Large bald patch (LP), (2) Medium bald patch (MP), (3) Small bald patch (SP), (4) Intact
grassland patch (IG), (5) above pika tunnel (PT) and (6) old Pika pile (PP).
**Figure 2.** Ecosystem respiration of different surface types: (1) large bald patch (LP), (2)
medium bald patch (MP), (3) small bald patch (SP), (4) intact grassland patch (IG), (5) above
pika tunnel (PT) and (6) old pika pile (PP). The upper solid lines, the bottom solid lines, the
bold solid horizontal line and the empty dot mean the maximum value, minimum value,
median and abnormal value. Letters on the error bars indicate significant differences among
different surface types at $P < 0.05$.
**Figure 3**. Daily average air temperature and precipitation of the study site in 2016.
**Figure 4.** Monthly average soil temperature and soil moisture at 10 cm depth under different
surface types: (1) large bald patch (LP), (2) medium bald patch (MP), (3) small bald patch
(SP), (4) intact grassland patch (IG), (5) above pika tunnel (PT) and (6) old pika pile (PP).
**Figure 5.**Soil saturated hydraulic conductivity (SHC) under different surface types: (1) large
bald patch (LP), (2) medium bald patch (MP), (3) small bald patch (SP), (4) intact grassland
patch (IG), (5) above pika tunnel (PT) and (6) old pika pile (PP).
**Figure 6. S**oil compactness under different surface types: (1) large bald patch (LP), (2)
medium bald patch (MP), (3) small bald patch (SP), (4) intact grassland patch (IG), (5) above
pika tunnel (PT) and (6) old pika pile (PP).
**Figure 7.** Soil organic carbon (SOC) (a) and total nitrogen (TN) (b) density of different
surface types: (1) large bald patch (LP), (2) medium bald patch (MP), (3) small bald patch
(SP), (4) intact grassland patch (IG), (5) above pika tunnel (PT) and (6) old pika pile (PP).
**Figure 8.** Aboveground biomass (AGB) (a) and belowground biomass (BGB) (b) under
different surface types: (1) large bald patch (LP), (2) medium bald patch (MP), (3) small bald
patch (SP), (4) intact grassland patch (IG), (5) above pika tunnel (PT) and (6) old pika pile
(PP).
**Figure 9**. The correlation coefficient charts between ecosystem respiration (Re) and biotic
and abiotic factors for all six land surfaces. The diagonal line in the figure shows the
distributions of the variables themselves. The red line means the frequency distribution of
variables. The lower triangle (the left bottom of the diagonal) in the figure shows scatter plots
of the two properties. The upper triangle (the upper right of the diagonal) in the figure
indicates the correlation values of the two parameters; the asterisk indicates the degree of
significance (*** indicates significant differences at $P < 0.001$, ** indicates significant
differences at $P < 0.01$, * indicates significant differences at $P < 0.05$.). The bold bigger
numbers mean the higher correlation.
**Figure 1.**

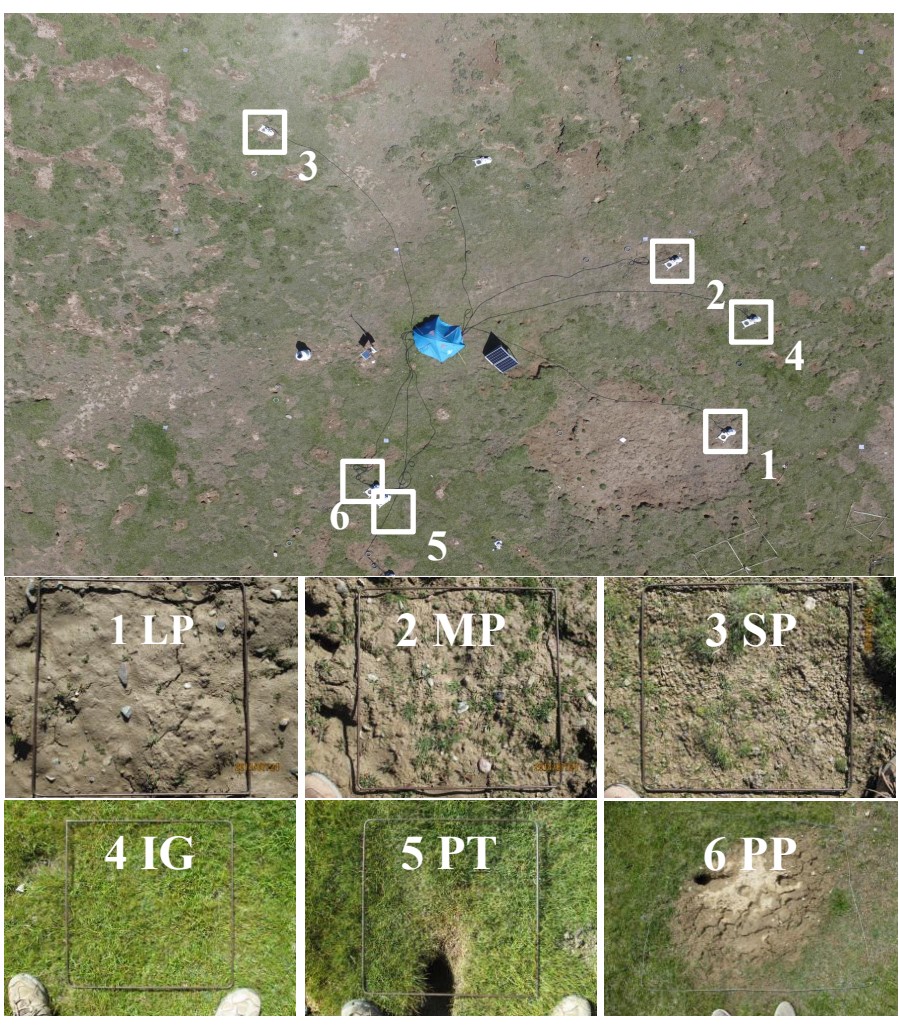

**Figure 2.**

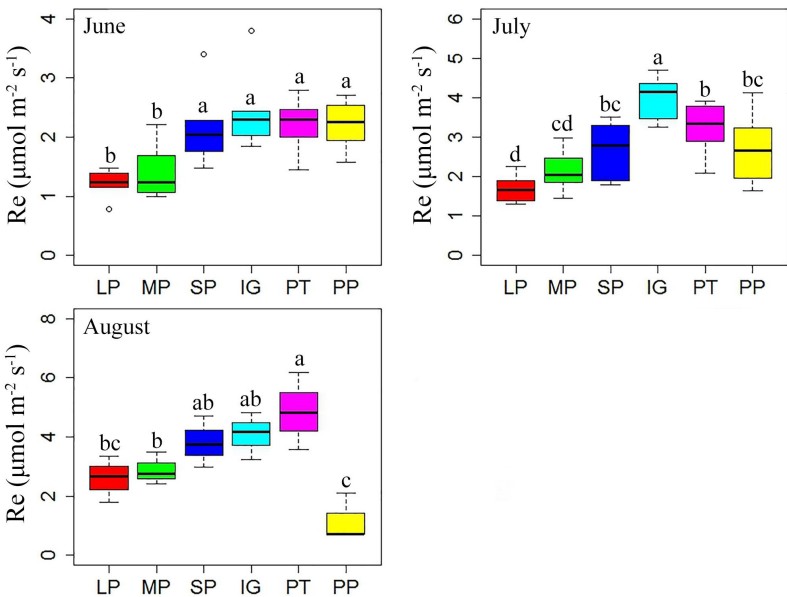



**Figure 3**.

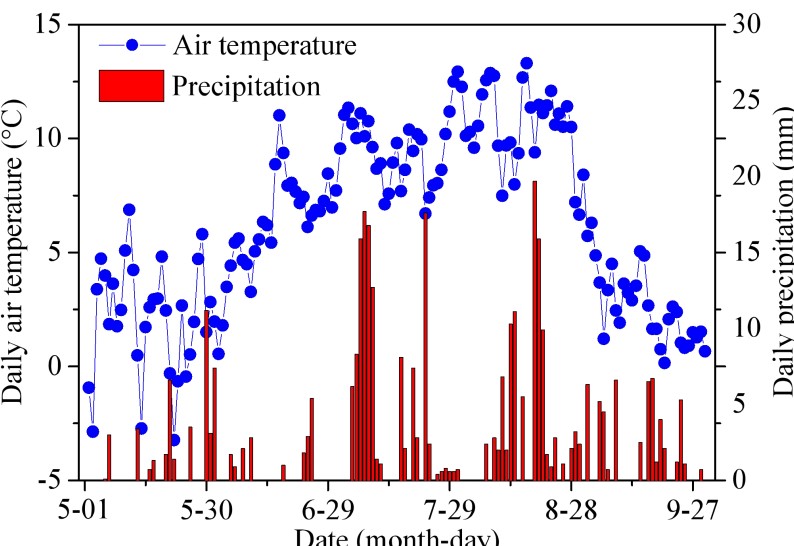


**Figure 4.**

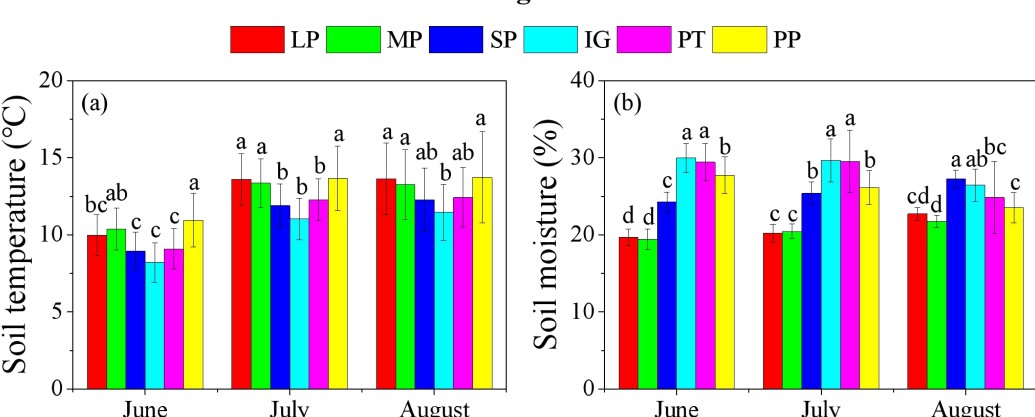


**Figure 5.**

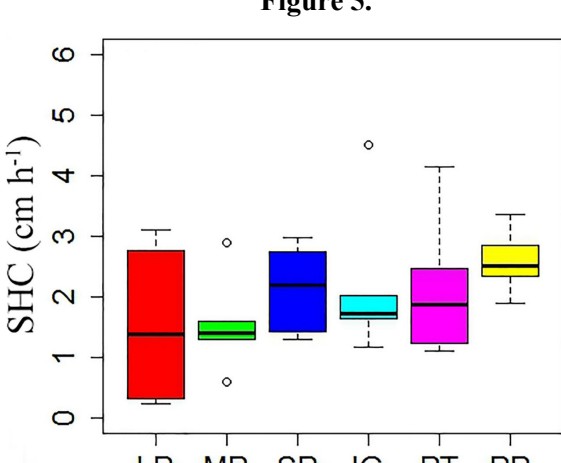


**Figure 6.**

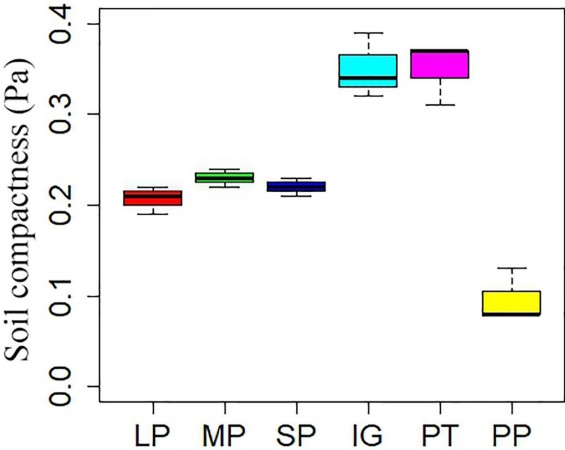


**Figure 7.**

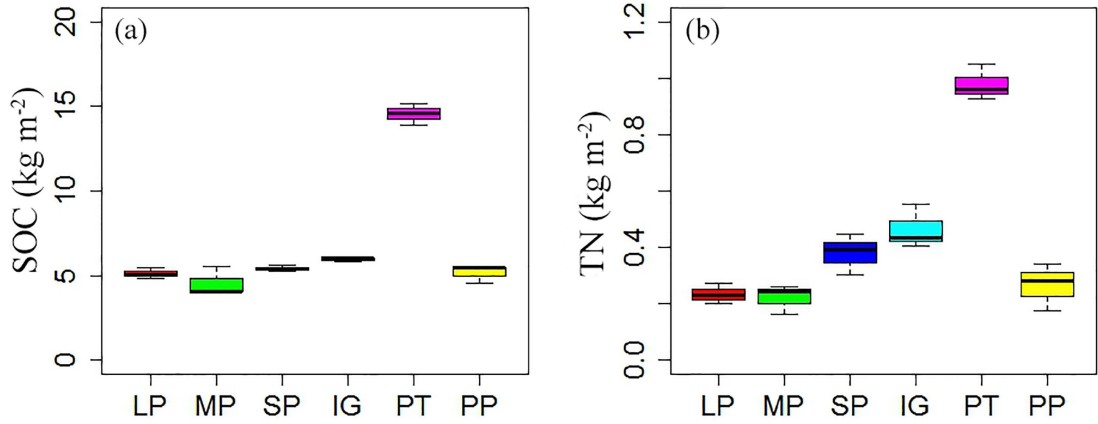


**Figure 8.**

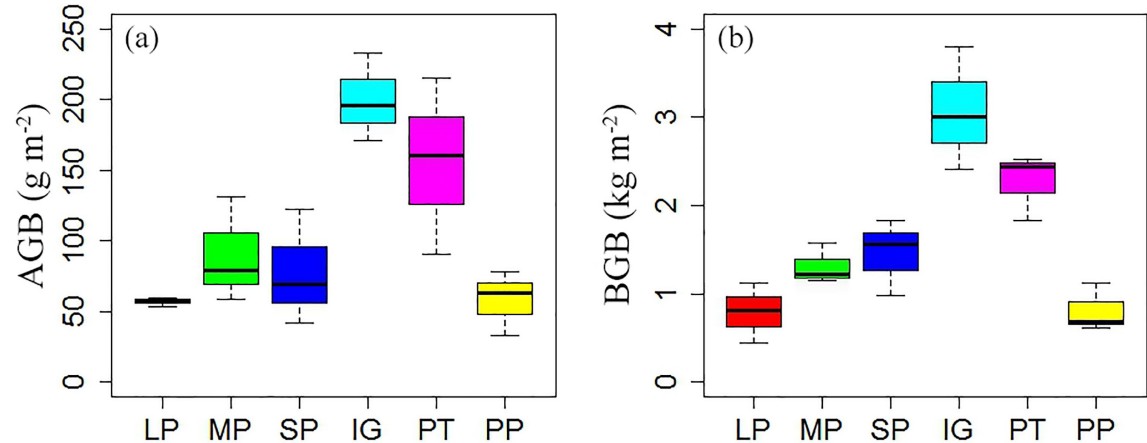


**Figure 9.**

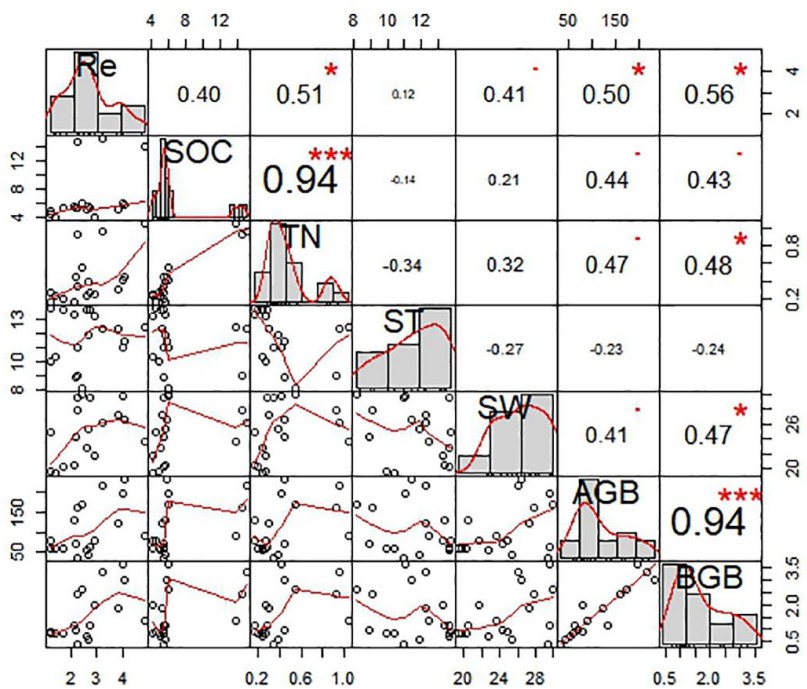
