# Peer review of "Effect of plateau pikas disturbance and patchiness on ecosystem carbon emission of"

_Biogeosciences, 2018_

## Referee Comment (RC1) · Anonymous Referee #1 · 5 Aug 2018

General comments: The plateau pika (Ochotona curzoniae) is one of the main native soil faunas on the Qinghai-Tibet Plateau and plays a key role in the terrestrial ecosystem there. Previous studies have mainly focused on its active habits and the influence of population density on soil properties, plant communities, and so on. On contrast, the present study aims to study the effect of plateau pika disturbance and patchiness on ecosystem carbon emission at the plot scale (i.e. large bald patch, medium bald patch, small bald patch, intact grassland, above pika tunnel and pika pile). The results are critical for ecological restoration and environmental change on the Qinghai-Tibetan Plateau.

[Figure]

Specific comments: (1) Introduction: This section has not clarified clearly why we should study the effect of plateau pika disturbance and patchiness on ecosystem carbon emission at the plot scale, but not at other scales? What are the exact differences between this study and so many previous studies? (2) Materials and methods: Line 114-118: Is there any standard to distinguish the six representative underlying surfaces? Especially how to determine the threshold area for the division of large, medium and small bald patches (i.e. 9 m2 and 1 m2)? Line 124-136: Were the soil temperature and moisture measured at all three 100 m × 100 m plots or only one 100 m × 100 m plots? Were the soil saturated hydraulic conductivity, soil hardness and ecosystem respiration rates measured for only one time or many times during the study periods? These key questions should be clarified. Line 138-141: How depth was the pika tunnel? Did this depth limit the collection of soil core to 40 cm? (3) Results: Line 180-182: The soil saturated hydraulic conductivity for all six surfaces types was very low. Please check the calculation process or data units. Line 203-204: Temperature is a mainly controlling factor of ecosystem respiration (line 50-51); however, regression analysis of this study showed that ecosystem respiration had no significant correlation with soil temperature. What is the reason for this unexpected result? (4) Discussion: Line 216-217: "Nevertheless, the increased water infiltration was unable to increase soil moisture under pika pile." Why? The potential reasons should be discussed. Line 227-229: The explanation for the low soil moisture under bald patches was not convincing, because the vegetation transpiration at intact grassland may be higher than the corresponding soil evaporation under bald patches at the same periods. Line 230-233: More details about the reason for the different soil temperature patterns should be added. Line 234-235: What is the reason for the description of "high soil temperature under bald patch was a disadvantage for the recovery of vegetation"?

Technical corrections: Line 33: Delete "under". Line 88-90: This sentence is not exact, because lots of previous researches have studied the heterogeneous underground vegetation and belowground soil properties. Line 188-189: This sentence has the same mean with the sentence in line 185-186. Line 197-198: According to the description in line 172, the growing season in the study is from May to September. Please add the data about ecosystem respiration in May and September. Line 214: Change "Figure 3" to "Figure 4". Line 311: Some references cited in the text were not listed in the "Reference" section. Line 518-520: Six small photos below the aerial photo are not clear. Moreover, add "MP" after "2". Line 539: The regression analysis was used to analyze the relationships of ecosystem respiration with biotic and abiotic factors (line 168-169). However, the result in figure 9 was only the correlation coefficient between them.

---

## Referee Comment (RC2) · Anonymous Referee #2 · 6 Aug 2018

This study aimed to address the impact of patchiness and pika disturbance on ecosystem respiration at an alpine meadow grassland. The topic is interesting and meaningful and they have presented a good dataset that is sufficient to address the questions they brought up. However, I think the storyline can be better organized and many technical details still need to be added. General comments: 1. According to the title of the article, the whole story should be centered on the ecosystem respiration. Therefore, I suggest the authors to re-organize the storyline by: (1) using the "intact grassland" type as a reference, which is the natural status of the site, and compare other types to IG to indicate the effects of patchiness or pika disturbance. (2) presenting the CO2 flux first, then environmental conditions and use the differences in soil conditions to explain the

flux differences. This applies to abstract, result, order of the figures and discussions. Particularly for discussion, consider separating the sections based on different effects (patchiness and pika disturbance) and explain what factors caused the difference in fluxes compared to the reference type (IG). 2. Method section needs to be expanded with more information on the details. See my comments on each specific line.

Specific comments: L52, other substrates? Such as? L57, ecological system? Ecosystem! L68, this definition of patchiness need to be referred to earlier in the paragraph. L89, not clear, others also studied the effect of pika disturbance and patchiness, which are what you meant as "heterogeneity" to my understanding. What makes your study different from theirs? L93, "underlying surface" sounds a little awkward. Change it to land surface or soil surface. Check this expression throughout the manuscript. L94, I think what you meant was "the spatial heterogeneity of Re" in aim (3). L105 "plant" species L121, according to your description, seems the fluxes were measured in different plots from ones that measured environmental conditions, right? If yes, how far away are they? Are they comparable? L126, "were" logged . . . L129, soil hardness is not a very familiar concept. Explain it and what unit is used? L131, since the respiration measurement is the key of this study, more details are needed. How big is the chamber? Transparent of opaque? How many replicates? Only one gas analyzer was used? How many minutes did one measurement take? What is the frequency of the data? During which period (specific dates) were the measurements taken? Also, how the fluxes were calculated? How the air temperature inside of the chamber was measured? L138 change "determined" to "collected". L142 from each surface type? L149 how many replicates? L150 change "sampled" to "determined" L152 each type? L169 according to your figure, this seems like correlation analysis instead of regression. Figure 2, which year? Average Ta? Figure 3, monthly average? Figure 8, $\mu$mol instead of umol Figure 9, this is not a good way to present correlation results. First, specify what analysis in the caption. Second, the full correlation table looks redundant as it presents two copies of each pair of variables. Also, correlation coefficients and P value need to be included. Was the correlation done across the different surface types?

---

## Author Comment (AC1) · 6 Oct 2018

Introduction: This section has not clarified clearly why we should study the effect of plateau pika disturbance and patchiness on ecosystem carbon emission at the plot scale, but not at other scales? What are the exact differences between this study and so many previous studies? Thank you for your careful review. We have revised the whole manuscript to eliminate the reader's confusion. In fact, we mainly focused on the effect of plateau pika disturbance and patchiness on ecosystem carbon emission of alpine meadow and we did not refer to any scales in this work. We have clarified clearly why we should study the effect of plateau pika disturbance and patchiness on

alpine grassland at plot scale in our previous study (Yi et al., 2016). Typically, most of the previous studies compared carbon fluxes under intact vegetation at plots with different number of pika burrows. However, ecosystem carbon emissions from the homogeneous land surface induced by pika piles and patchiness have yet to be quantified. These are the exact differences between this study and so many previous studies. Yi, S., Chen, J., Qin, Y., Xu, G.: The burying and grazing effects of plateau pika on alpine grassland are small: a pilot study in a semiarid basin on the Qinghai-Tibet Plateau, Biogeosciences, 13(22), 6273-6284, 2016. Materials and methods: Line 114-118: Is there any standard to distinguish the six representative underlying surfaces? Especially how to determine the threshold area for the division of large, medium and small bald patches (i.e. 9 m2 and 1 m2)? Thank you for your careful review. Six representative underlying surfaces were selected according to the previous work in our study site (Yi et al., 2016; Qin et al., 2018). They were distinguished easily in aerial photographs. Large bald patches had less vegetation cover and the smallest side was larger than 3 m. Medium patches also covered by less vegetation cover and the larger side was in a range of 1 to 3 m and small bald patches were characteristic by less vegetation cover and the larger side was less than 1 m. Intact grassland was characteristic by high vegetation cover and no large and medium bare land was found. Pika tunnel and pika pile usually co-exited. Pika tunnel is approximately 6 cm in diameter and pika pile is in the front of pika tunnel, 60 cm in diameter and less vegetation cover. We calculated the threshold area of large, medium and small patches by aerial photograph. Each aerial photograph has 12 million pixels. At a height of 20 m, the resolution of each pixel is ∼1 cm and each photograph covers ∼26 m × 35 m of ground. Pixels in each aerial image were first classified into two groups, i.e. vegetated or bare patches (Yi, 2016). Then patches with different sizes were created using OpenCv Library. And finally, fractions of vegetation and bare patches (large, medium and small patches) were calculated. We revised this part as follow (Line 114-118). "At early June 2016, three 100 m × 100 m plots were established as replicates. In each plot, six representative land surfaces were selected: (1) large bald patch with size larger than 9.0 m2 (LP), (2) medium bald patch with size of 1.0-9.0 m2 (MP), (3) small bald patch with size of less than 1.0 m2 (SP), (4) intact grassland patch (IG), (5) above pika tunnel (PT), (6) old pika pile (PP) (Figure 1) (Yi et al., 2016; Qin et al., 2018)." Yi, S.H., 2016. FragMAP: a tool for long-term and cooperative monitoring and analysis of small-scale habitat fragmentation using an unmanned aerial vehicle. Int. J. Remote Sens. 1-12. http://dx.doi.org/10.1080/01431161.2016.1253898. Yi, S., Chen, J., Qin, Y., Xu, G.: The burying and grazing effects of plateau pika on alpine grassland are small: a pilot study in a semiarid basin on the Qinghai-Tibet Plateau, Biogeosciences, 13(22), 6273-6284, 2016. Qin, Y., Yi, S., Ding, Y., Xu, G., Chen, J., Wang, Z.: Effects of small-scale patchiness of alpine grassland on ecosystem carbon and nitrogen accumulation and estimation in northeastern qinghai-tibetan plateau, Geoderma, 318, 52-63, 2018. Line 124-136: Were the soil temperature and moisture measured at all three 100 m $\times$ 100 m plots or only one 100 m $\times$ 100 m plots? Thank you for your question. Soil temperature and moisture were measured in one 100 m $\times$ 100 m plot where ecosystem respiration was measured. Both soil temperature and moisture were measured with three replicates under each underlying surface type. We revised this part to eliminate the confusion (Line 124-127). "Soil temperature and moisture at 10 cm were measured in a 100 m $\times$ 100 m plot where ecosystem respiration was measured by using an auto-measurement system (Decagon Inc., USA) from early June to the late August. The system consisted of an EM50 logger and five 5TM sensors. The data were logged automatically every 30 min" Were the soil saturated hydraulic conductivity, soil hardness and ecosystem respiration rates measured for only one time or many times during the study periods? These key questions should be clarified. Thanks for your suggestion. Soil saturated hydraulic conductivity and soil hardness under each surface type were measured one time every month from June to August. Ecosystem respiration was measured every 7-10 days from June 16 to August 20 depending on weather conditions. We therefore revised this part as follow (Line 124-155). "Soil temperature and moisture at 10 cm were measured in a 100 m $\times$ 100 m plot where ecosystem respiration was measured by using an auto-measurement system (Decagon Inc., USA)

from early June to the late August. The system consisted of an EM50 logger and five 5TM sensors. The data were logged automatically every 30 min. Soil saturated hydraulic conductivity and compactness were measured once each month from June to August. Soil saturated hydraulic conductivity was measured by Dual Head infiltrometer (Decagon Inc., USA). The measurement process included 15 min soak time, 20 min hold time at low pressure head (5 cm) and high pressure head (15 cm) with 2 cycles. Each measurement takes 95 min altogether. Soil compactness was measured with TJSD-750 (Hangzhou Top Instrument co., LTD, Hangzhou, China) from the soil surface to 10 cm depth. Ecosystem respiration rates were measured using the LICOR-8150 Automated Soil CO2 Flux System, which was an accessory for the LI-8100A with at most 8 individual chambers at one time. Ecosystem CO2 emission was sampled and controlled by the LI-8100A Analyzer Control Unit. The air temperature inside of the chamber was measured using the internal thermistor of the chamber. The ecosystem CO2 fluxes were calculated by the equation as follow.

where Fc is the soil CO2 efflux rate ($\mu$mol m-2 s-1), V is volume (cm3), P0 is the initial pressure (kPa), W0 is the initial water vapor mole fraction (mmol mol-1), S is soil surface area (cm2), T0 is initial air temperature ($^\circ$C), and $\partial$C'/$\partial$t is the initial rate of change in water-corrected CO2 mole fraction ($\mu$mol-1 mol s-1). Six LICOR-8100-104 long-term opaque chambers (20cm in diameter LICOR, Inc., Lincoln, NE, USA) were used to measure alternately between three replicates for six land surface types. Therefore, 3 days at least were required to complete one rotation measurements of ecosystem respiration. To measure ecosystem respiration, eighteen polyvinyl chloride collars with a 20 cm inner diameter and a 12 cm height were inserted into the soil with 3-4 cm exposed to the air (Qin et al., 2013). All of the collars were installed at least 24 h before the first measurement to reduce disturbance-induced ecosystem CO2 effluxes. Ecosystem respiration rates were measured every 7-10 days from June 16 to August 20 in 2016 depending on weather conditions. A round-the-clock measurement protocol was carried out and ecosystem respiration rates were measured every 30 minutes. Each measurement takes 1 minute and 45 seconds, including pre-purge seconds, dead band 15 seconds, observation length 1 minute and post-purge 20 seconds." Line 138-141: How depth was the pika tunnel? Did this depth limit the collection of soil core to 40 cm? Thanks for your question. We investigated pika tunnel by digging soil pole and the depth of pika tunnel was about 40cm. Therefore, it wasn't difficulty to collect soil core at depth of 40cm. We have revised this part as follow (Line 157-164). "Soil samples were collected during the periods of late July to early August 2016. In each surface type of each plot, five soil cores were collected using a stainless-steel auger (5 cm in diameter) at depths of 0-10, 10-20, 20-30 and 30-40 cm, and bulked as one composite sample for each depth in each quadrat. Another five soil cores were sampled by cylindrical cutting ring (7 cm in diameter and 5.2 cm in depth) to determine soil bulk density from each land surface type. Pika tunnel was approximate 6 cm in diameter and 40 cm in depth. Therefore, soil samples were available to collect at depth of 40cm. Totally, 512 soil samples were collected." Discussion: Line 216-217: "Nevertheless, the increased water infiltration was unable to increase soil moisture under pika pile." Why? The potential reasons should be discussed. Thanks for your suggestion. We discussed the reason why the increased water infiltration was unable to increase soil moisture under pika pile as follow (Line 263-271). "Nevertheless, the increased water infiltration was unable to increase soil moisture under pika piles. For example, soil moisture under pika piles was approximate 5 % lower than intact grassland (Figure 4). Our result was discrepant with previous studies which reported old pika mound had the highest soil moisture during the summer (Ma et al., 2018) and moderate pika burrowing activities increased surface soil moisture (Li and Zhang, 2006). This difference may be contributed to the high pika density in alpine meadow (Guo et al, 2017). Moreover, pika piles were loose (Figure 6) with less vegetation cover (Figure 8), which was not beneficial for soil moisture storage." Line 227-229: The explanation for the low soil moisture under bald patches was not convincing, because the vegetation transpiration at intact grassland may be higher than the corresponding soil evaporation under bald patches at the same periods. Thanks for your comment. In fact, we have measured evaporation under different surfaces of the intact grassland, isolate grassland, large patches, medium patches and small patches since the early June 2016. It is difficult to measure evaporation from pika tunnle and pika pile due to their small sizes. Therefore, these data were not presented in this manuscript. We found that the evaporation under bald patches were higher than the intact grassland in our study sites through three years observation. We have revised this part as follow (Line 288-297). "Our results showed that soil moisture under large and medium patches decreased 10 % than intact grassland (Figure 4). Previous studies had reported that the soil compaction of bald patches decreased the rate of water infiltration (Wuest et al., 2006; Wilson and Smith, 2015), which was similar with our results showed that bald patches had less saturated soil hydraulic conductivity (Figure 5). Low vegetation cover under bald patches was not beneficial for water retention and utilization, where most of soil water was mainly lost as a way of evaporation (Yi et al., 2014). We have measured evaporation of the intact grassland, isolate grassland, large patches, medium patches and small patches since the early June 2016. Three years results indicated that evaporation under bald patches were higher than the intact grassland (data were not shown here)." Line 230-233: More details about the reason for the different soil temperature patterns should be added. Thank you for your suggestion. We have added more detailed information about the difference of soil temperature between intact grassland and pika pile and bald patches. This part has been revised as follow (Line 301-309). "Our results indicated that soil temperature under pika piles and bald patches was approximate 1 to 3 °C higher than intact grassland (Figure 4), which mainly resulted from the heterogeneity of surface albedo, surface soil water retention, heat conduction properties and radiation (Beringer et al., 2005; Pielke, 2005; Yi et al., 2013; You et al., 2017). It was suggested that pikas disturbance create a better soil temperature buffer for them to avoid the extreme cold in winter (Ma et al., 2018), whereas high soil temperature under bald patch was a disadvantage for the recovery of vegetation because patch surface had the smallest soil moisture content (Figure 4) and the largest daily range of soil temperature (Ma et al., 2018)." Line 234-235:
What is the reason for the description of "high soil temperature under bald patch was a disadvantage for the recovery of vegetation"? Thank you for your question. Our study site belongs to semi-arid region, where water was one of dominant limit factors for vegetation growth. Patch surface had the smallest soil moisture content and the largest daily range of soil temperature, which was not beneficial for soil water retention. We have changed this part as follow (Line 305-309). "It was suggested that pikas disturbance create a better soil temperature buffer for them to avoid the extreme cold in winter (Ma et al., 2018), whereas high soil temperature under bald patch was a disadvantage for the recovery of vegetation because patch surface had the smallest soil moisture content (Figure 4) and the largest daily range of soil temperature (Ma et al., 2018)." Ma, Y.J., Wu, Y.N., Liu, W.L., Li, X.Y., Lin, H.S.: Microclimate response of soil to plateau pika's disturbance in the northeast qinghai-tibet plateau, European Journal of Soil Science, 69(2), 232-244, 2018. Technical corrections: Line 33: Delete "under". Thank you for your suggestion. We have deleted "under" according to your suggestion. Line 88-90: This sentence is not exact, because lots of previous researches have studied the heterogeneous underground vegetation and belowground soil properties. Thank you for your suggestion. We totally agree with your comment that lots of previous researches have studied the heterogeneous underground vegetation and belowground soil properties. However, few studies have investigated the difference of ecosystem respiration under the heterogeneous underlying surface. Therefore, we have changed this sentence to "Nevertheless, most of these studies have mainly focused on ecosystem carbon emission rate under the homogeneous land surface rather than heterogeneous land surfaces." (Line 88-90) Line 188-189: This sentence has the same mean with the sentence in line 185-186. Thank you for your suggestion. We have deleted this sentence according to your suggestion. Line 197-198: According to the description in line 172, the growing season in the study is from May to September. Please add the data about ecosystem respiration in May and September. Thank you for your suggestion. Actually, our field observation started at the early June and finished at the late August in 2016. It's pity we can't add the
data of ecosystem respiration in May and September. Line 214: Change "Figure 3" to "Figure 4". Thank you for your suggestion. We have changed "Figure 3" to "Figure 4" according to your suggestion. Line 311: Some references cited in the text were not listed in the "Reference" section. Thank you for your suggestion. The references have been checked carefully through manuscript according to your suggestion. And now all the references cited in the manuscript are also included in the "Reference" section. Line 518-520: Six small photos below the aerial photo are not clear. Moreover, add "MP" after "2". Thank you for your suggestion. We have redrawn Figure 1 according to your suggestion. We also add "MP" after "2". We believe that the photos are clear now. Line 539: The regression analysis was used to analyze the relationships of ecosystem respiration with biotic and abiotic factors (line 168-169). However, the result in figure 9 was only the correlation coefficient between them. Thank you for your suggestion. We have redrawn Figure 9 according to your suggestion and now it contained both the correlation coefficients and P value in one figure. The title of Figure 9 was changed to "Figure 9. The correlation coefficient charts between ecosystem respiration (Re) and biotic and abiotic factors for all six land surfaces. The diagonal line in the figure shows the distributions of the variables themselves. The lower triangle (the left bottom of the diagonal) in the figure shows scatter plots of the two properties. The upper triangle (the upper right of the diagonal) in the figure indicates the correlation values of the two parameters; the asterisk indicates the degree of significance (*** indicates significant differences at P < 0.001, * indicates significant differences at P < 0.01, * indicates significant differences at P < 0.05.). The bold bigger numbers mean the higher correlation."

Please also note the supplement to this comment:
https://www.biogeosciences-discuss.net/bg-2018-296/bg-2018-296-AC1-supplement.pdf

**Supplement:**

[revised manuscript text omitted]

---

## Author Comment (AC2) · 6 Oct 2018

This study aimed to address the impact of patchiness and pika disturbance on ecosystem respiration at an alpine meadow grassland. The topic is interesting and meaningful and they have presented a good dataset that is sufficient to address the questions they brought up. However, I think the storyline can be better organized and many technical details still need to be added. General comments: 1. According to the title of the article, the whole story should be centered on the ecosystem respiration. Therefore, I suggest the authors to re-organize the storyline by: (1) using the "intact grassland" type as a reference, which is the natural status of the site, and compare other types to IG to

indicate the effects of patchiness or pika disturbance. (2) presenting the CO2 flux first, then environmental conditions and use the differences in soil conditions to explain the flux differences. This applies to abstract, result, order of the figures and discussions. Particularly for discussion, consider separating the sections based on different effects (patchiness and pika disturbance) and explain what factors caused the difference in fluxes compared to the reference type (IG). 2. Method section needs to be expanded with more information on the details. See my comments on each specific line. Thank you for your suggestion. The storyline were re-organized and the whole manuscript has been revised according to your suggestion. Abstract (Line 21-41) "
[revised manuscript text omitted]
., 2016). " Specific comments: L52, other substrates? Such as? Thank you for your question. The substrates affected ecosystem respiration included carbohydrate fixed by leaves, vegetation litter and soil organic matter. We have revised the manuscript as follow (Line 49-53). "Dependent on autotrophic (plant) and heterotrophic (microbe) activity, ecosystem respiration is mainly controlled by abiotic factors (primarily temperature and water availability) (Chimner and Welker, 2005; Flanagan and Johnson, 2005; Nakano et al., 2008; Buttlar et al., 2018), and supply of carbohydrate fixed by leaves, vegetation litter and soil organic matter (Janssens et al., 2001; Reichstein et al., 2002)." L57, ecological system? Ecosystem! Thank you for your suggestion. We have changed ecological system to ecosystem according to your suggestion (Line 57). L68, this definition of patchiness need to be referred to earlier in the paragraph. Thank you for your suggestion. The definition of patchiness has been moved to earlier in the paragraph according to your suggestion. We revised this part as follow (Line 56-77). "One of the basic function of terrestrial ecosystem is to regulate carbon balance between the atmosphere and ecosystem (Canadell et al., 2007; Le Quéré et al., 2014; Ahlström et al., 2015). However, this balance would be broken by widespread land degradation (Post and Kwon, 2000; Dregne, 2002), which accompanied with the reduction of photosynthetic fixed carbon dioxide from atmosphere and carbon sequestration by soils (Defries et al., 1999; Upadhyay et al., 2005). It was estimated that land degradation had resulted in 19-29 Pg C loss worldwide (Lal, 2001). Over the past decades, grasslands have experienced patchiness throughout the world and this process is still ongoing (Baldi et al., 2006; Wang et al., 2009; Roch and Jaeger, 2014). Patchiness generally refers to a landscape that consists of remnant areas of native vegetation surrounded by a more heterogeneous and patchy situation (Kouki and Löfman, 1998). Other than climate change (Yi et al., 2014), vegetation self-organization (Rietkerk et al., 2004; Venegas et al., 2005; McKey et al., 2010) or anthropogenic disturbances (Kouki and Löfman, 1998; Yi et al., 2016), rodents burrowing activities were also considered as the origin of the patchiness (Wei et al., 2007; Davidson and Lightfoot, 2008). This patchiness intensified spatial heterogeneity of land surface and led to the changing of the structure and function of the original ecosystem (Herkert et al., 2003; Bestelmeyer et al., 2006; Lindenmayer and Fischer, 2013). For instance, there is abundant evidence that patchiness not only intensified the spatial heterogeneous distribution of ecosystem organic carbon (C) and vegetation productivity (Yan et al., 2016; Qin et al., 2018) but also altered the pattern of coupled water and heat cycling between the land surface and the atmosphere (Saunders et al., 1991; You et al., 2017; Ma et al., 2018). Consequently, this may alter ecosystem carbon emission process (Juszczak et al., 2013)." L89, not clear, others also studied the effect of pika disturbance and patchiness, which are what you meant as "heterogeneity" to my understanding. What makes your study different from theirs? Thank you for your question. We totally agree with your comment that lots of previous researches have studied the heterogeneous underground vegetation and belowground soil properties. However, few studies have investigated the difference of ecosystem respiration under the heterogeneous underlying surface.
Here we mainly meant the heterogeneity of ecosystem respiration. Therefore, we have changed this sentence to "Nevertheless, most of these studies have mainly focused on ecosystem carbon emission rate under the homogeneous land surface rather than heterogeneous land surfaces." Typically, most of the previous studies compared carbon fluxes under intact vegetation at plots with different number of pika burrows. However, ecosystem carbon emissions from the heterogeneous land surface induced by pika piles and patchiness have yet to be quantified. These are the exact differences between this study and so many previous studies. L93, "underlying surface" sounds a little awkward. Change it to land surface or soil surface. Check this expression throughout the manuscript. Thank you for your suggestion. We have changed "underlying surface" to "land surface" in the whole manuscript according to your suggestion. L94, I think what you meant was "the spatial heterogeneity of Re" in aim. Thank you for your suggestion. We have revised the third aim according to your suggestion. We have revised the manuscript as follow (Line 92-95). "Thus, the specific aims of this study were to (1) investigate the spatial heterogeneity of Re under the effect of pikas and patchiness; (2) illuminate the potential regulating mechanism of pikas disturbance and patchiness to ecosystem respiration (Re) in an alpine meadow grassland in the northeastern part of Qinghai-Tibetan Plateau (QTP)." L105 "plant" species Thank you for your suggestion. We have changed "species" to "plant species" according to your suggestion. L121, according to your description, seems the fluxes were measured in different plots from ones that measured environmental conditions, right? If yes, how far away are they? Are they comparable? Thank you for your question. Ecosystem respiration, soil temperature and moisture were measured in one $100 \times 100$ m plot and with three replicates under each land surface. Soil and vegetation were measured in all three $100 \times 100$ m plots. Each $100 \times 100$ m plot was in a distance of less than 50 m, which has the similar plant and terrain. We therefore believed they were comparable. L126, "were" logged . . . Thank you for your suggestion. We have changed "The Data logged automatically every 30 min" to "The data were logged automatically every 30 min" according to your suggestion. L129, soil hardness is not a very familiar concept.

Explain it and what unit is used? Thank you for your suggestion. We have changed "soil hardness" to "soil compactness" according to your suggestion. We also added it unit both in result and Figure 5. "Soil compactness was over 0.30 Pa in intact grassland patch and above pika tunnel, approximate 0.20 Pa for bald patches and less than 0.10 Pa for pika pile (Figure 5), respectively. " L131, since the respiration measurement is the key of this study, more details are needed. How big is the chamber? Transparent of opaque? How many replicates? Only one gas analyzer was used? How many minutes did one measurement take? What is the frequency of the data? During which period (specific dates) were the measurements taken? Also, how the fluxes were calculated? How the air temperature inside of the chamber was measured? Thank you for your suggestion. We have added more information regarding ecosystem respiration measurement according to your suggestion (Line 133-155). "Ecosystem respiration rates were measured using the LICOR-8150 Automated Soil $CO_2$ Flux System, which was an accessory for the LI-8100A with at most 8 individual chambers at one time. Ecosystem $CO_2$ emission was sampled and controlled by the LI-8100A Analyzer Control Unit. The air temperature inside of the chamber was measured using the internal thermistor of the chamber. The ecosystem $CO_2$ fluxes were calculated by the equation as follow.

where $F_c$ is the soil $CO_2$ efflux rate ($\mu$mol m-2 s-1), V is volume (cm3), $P_0$ is the initial pressure (kPa), $W_0$ is the initial water vapor mole fraction (mmol mol-1), S is soil surface area (cm2), $T_0$ is initial air temperature (°C), and $\partial C'/\partial t$ is the initial rate of change in water-corrected $CO_2$ mole fraction ($\mu$mol-1 mol s-1). Six LICOR-8100-104 long-term opaque chambers (20cm in diameter LICOR, Inc., Lincoln, NE, USA) were used to measure alternately between three replicates for six land surface types. Therefore, 3 days at least were required to complete one rotation measurements of ecosystem respiration. To measure ecosystem respiration, eighteen polyvinyl chloride collars with a 20 cm inner diameter and a 12 cm height were inserted into the soil with 3-4 cm exposed to the air (Qin et al., 2013). All of the collars were installed at least 24 h before the first measurement to reduce disturbance-induced ecosystem $CO_2$ effluxes. Ecosystem respiration rates were measured every 7-10 days from June 16 to

August 20 in 2016 depending on weather conditions. A round-the-clock measurement protocol was carried out and ecosystem respiration rates were measured every 30 minutes. Each measurement takes 1 minute and 45 seconds, including pre-purge 10 seconds, dead band 15 seconds, observation length 1 minute and post-purge 20 seconds." L138 change "determined" to "collected". Thank you for your suggestion. We have changed "determined" to "collected" according to your suggestion. L142 from each surface type? Thank you for your careful review. The sentence has changed to "Another five soil cores were sampled by cylindrical cutting ring (7 cm in diameter and 5.2 cm in depth) to determine soil bulk density from each land surface type." according to your suggestion. L149 how many replicates? Thank you for your careful review. Soil and vegetation samples were collected under six land surface types with three replicates in three 100 × 100 m plots. To eliminate the confusion, we have revised this part as follow (Line 171-176). "There were a total of 108 aboveground and belowground vegetation samples (3 plots × 6 land surface types × 3 replicates) from the study area. Aboveground biomass was determined by clipping all above-ground living plants at ground level, drying (oven-dried at 65°C for 48 h) and weighing. Belowground biomass was sampled by collecting five soil columns, and each soil column was 5 cm in diameter and 40 cm in depth." L150 change "sampled" to "determined" Thank you for your careful review. We have changed "sampled" to "determined" according to your suggestion (Line 173). L152 each type? Thank you for your careful review. It means each soil columns. To eliminate the confusion, this sentence was changed to "There were a total of 108 aboveground and belowground vegetation samples (3 plots × 6 land surface types × 3 replicates) from the study area. Aboveground biomass was determined by clipping all above-ground living plants at ground level, drying (oven-dried at 65°C for 48 h) and weighing. Belowground biomass was sampled by collecting five soil columns, and each soil column was 5 cm in diameter and 40 cm in depth."(Line 171-176) L169, according to your figure, this seems like correlation analysis instead of regression. Thank you for your careful review. We have changed "regression analysis" to "correlation analysis" according to your suggestion. Figure 2, which year? Average

Ta? Thank you for your careful review. All data in this manuscript were collected in 2016. Ta was daily average air temperature. To eliminate confusion, the title of Figure 2 has been changed to "Figure 2. Daily average air temperature and precipitation of the study site in 2016." Figure 3, monthly average? Thank you for your question. Both soil temperature and soil moisture were monthly average. To eliminate confusion, the title of Figure 3 has been changed to "Figure 3. Monthly average soil temperature and soil moisture under different surface types: (1) large bald patch (LP), (2) medium bald patch (MP), (3) small bald patch (SP), (4) intact grassland patch (IG), (5) above pika tunnel (PT) and (6) old pika pile (PP)." Figure 8, $\mu$mol instead of umol Thank you for your suggestion. We have replaced "umol" by "$\mu$mol" according to your suggestion. Figure 9, this is not a good way to present correlation results. First, specify what analysis in the caption. Second, the full correlation table looks redundant as it presents two copies of each pair of variables. Also, correlation coefficients and P value need to be included. Was the correlation done across the different surface types? Thank you for your suggestion. We have redrawn Figure 9 according to your suggestion. And now it contained both the correlation coefficients and P value in one figure. The correlation of ecosystem respiration with biotic and abiotic factors were done across the different surface types. The title of Figure 9 was changed to "Figure 9. The correlation coefficient charts between ecosystem respiration (Re) and biotic and abiotic factors for all six land surfaces. The diagonal line in the figure shows the distributions of the variables themselves. The lower triangle (the left bottom of the diagonal) in the figure shows scatter plots of the two properties. The upper triangle (the upper right of the diagonal) in the figure indicates the correlation values of the two parameters; the asterisk indicates the degree of significance (*** indicates significant differences at P < 0.001, * indicates significant differences at P < 0.01, * indicates significant differences at P < 0.05.). The bold bigger numbers mean the higher correlation."

The revised manuscript has been resubmitted to the Biogeosciences. We look forward to your decision.

[Figure]

Please also note the supplement to this comment:
https://www.biogeosciences-discuss.net/bg-2018-296/bg-2018-296-AC2-
supplement.pdf

**Supplement:**

[revised manuscript text omitted]

---

## Author Comment (AC3) · 16 Oct 2018

The comment was uploaded in the form of a supplement.
* * *

---

## Author Comment (AC4) · 16 Oct 2018

Response to Anonymous Referee #2's comments:

General comments: This study aimed to address the impact of patchiness and pika disturbance on ecosystem respiration at an alpine meadow grassland. The topic is interesting and meaningful and they have presented a good dataset that is sufficient to address the questions they brought up. However, I think the storyline can be better organized and many technical details still need to be added. General comments: 1. According to the title of the article, the whole story should be centered on the ecosystem respiration. Therefore, I suggest the authors to re-organize the storyline by: (1) using the "intact grassland" type as a reference, which is the natural status of the site, and compare other types to IG to indicate the effects of patchiness or pika disturbance. (2) presenting the $CO_2$ flux first, then environmental conditions and use the differences in soil conditions to explain the flux differences. This applies to abstract, result, order of the figures and discussions. Particularly for discussion, consider separating the sections based on different effects (patchiness and pika disturbance) and explain what factors caused the difference in fluxes compared to the reference type (IG). 2. Method section needs to be expanded with more information on the details. See my comments on each specific line.

Our reply: Thank you for your suggestion. The storyline were re-organized and the whole manuscript has been revised according to your suggestion in the section of abstract, result, order of the figures and discussions.

**Abstract** (Line 21-41)

[revised manuscript text omitted]

Specific comments:

(1) L52, other substrates? Such as?

Our reply: Thank you for your question. The substrates affected ecosystem respiration included carbohydrate fixed by leaves, vegetation litter and soil organic matter. We have revised the manuscript as follow (Line 49-53).

"Dependent on autotrophic (plant) and heterotrophic (microbe) activity, ecosystem respiration is mainly controlled by abiotic factors (primarily temperature and water availability) (Chimner and Welker, 2005; Flanagan and Johnson, 2005; Nakano et al., 2008; Buttlar et al., 2018), and supply of carbohydrate fixed by leaves, vegetation litter and soil organic matter (Janssens et al., 2001; Reichstein et al., 2002)."

(2) L57, ecological system? Ecosystem!

Our reply: Thank you for your suggestion. We have changed ecological system to ecosystem according to your suggestion (Line 57).

(3) L68, this definition of patchiness need to be referred to earlier in the paragraph.

Our reply: Thank you for your suggestion. The definition of patchiness has been moved to earlier in the paragraph according to your suggestion. We revised this part as follow (Line 56-77).

"One of the basic function of terrestrial ecosystem is to regulate carbon balance between the atmosphere and ecosystem (Canadell et al., 2007; Le Quéré et al., 2014; Ahlström et al., 2015). However, this balance would be broken by widespread land degradation (Post and Kwon, 2000; Dregne, 2002), which accompanied with the reduction of photosynthetic fixed carbon dioxide from atmosphere and carbon sequestration by soils (Defries et al., 1999; Upadhyay et al., 2005). It was estimated that land degradation had resulted in 19-29 Pg C loss worldwide (Lal, 2001). Over the past decades, grasslands have experienced patchiness throughout the world and this process is still ongoing (Baldi et al., 2006; Wang et al., 2009; Roch and Jaeger, 2014). Patchiness generally refers to a landscape that consists of remnant areas of native vegetation surrounded by a more heterogeneous and patchy situation (Kouki and Löfman, 1998). Other than climate change (Yi et al., 2014), vegetation self-organization (Rietkerk et al., 2004; Venegas et al., 2005; McKey et al., 2010) or anthropogenic disturbances (Kouki and Löfman, 1998; Yi et al., 2016), rodents burrowing activities were also considered as the origin of the patchiness (Wei et al., 2007; Davidson and Lightfoot, 2008). This patchiness intensified spatial heterogeneity of land surface and led to the changing of the structure and function of the original ecosystem (Herkert et al., 2003; Bestelmeyer et al., 2006; Lindenmayer and Fischer, 2013). For instance, there is abundant evidence that patchiness not only intensified the spatial heterogeneous distribution of ecosystem organic carbon (C) and vegetation productivity (Yan et al., 2016; Qin et al., 2018) but also altered the pattern of coupled water and heat cycling between the land surface and the atmosphere (Saunders et al., 1991; You et al., 2017; Ma et al., 2018). Consequently, this may alter ecosystem carbon emission process (Juszczak et al., 2013)."

(4) L89, not clear, others also studied the effect of pika disturbance and patchiness, which are what you meant as "heterogeneity" to my understanding. What makes your study different from theirs?

Our reply: Thank you for your question. We totally agree with your comment that lots of previous researches have studied the heterogeneous underground vegetation and belowground soil properties. However, few studies have investigated the difference of ecosystem respiration under the heterogeneous underlying surface. Here we mainly meant the heterogeneity of ecosystem respiration. Therefore, we have changed this sentence to "Nevertheless, most of these studies have mainly focused on ecosystem carbon emission rate under the homogeneous land surface rather than heterogeneous land surfaces."

Typically, most of the previous studies compared carbon fluxes under intact vegetation at plots with different number of pika burrows. However, ecosystem carbon emissions from the heterogeneous land surface induced by pika piles and patchiness have yet to be quantified. These are the exact differences between this study and so many previous studies.

(5) L93, "underlying surface" sounds a little awkward. Change it to land surface or soil surface. Check this expression throughout the manuscript.

Our reply: Thank you for your suggestion. We have changed "underlying surface" to "land surface" in the whole manuscript according to your suggestion.

(6) L94, I think what you meant was "the spatial heterogeneity of Re" in aim.

Our reply: Thank you for your suggestion. We have revised the third aim according to your suggestion. We have revised the manuscript as follow (Line 92-95).

"Thus, the specific aims of this study were to (1) investigate the spatial heterogeneity of Re under the effect of pikas and patchiness; (2) illuminate the potential regulating mechanism of pikas disturbance and patchiness to ecosystem respiration (Re) in an alpine meadow grassland in the northeastern part of Qinghai-Tibetan Plateau (QTP)."

(7) L105 "plant" species

Our reply: Thank you for your suggestion. We have changed "species" to "plant species" according to your suggestion.

(8) L121, according to your description, seems the fluxes were measured in different plots from ones that measured environmental conditions, right? If yes, how far away are they? Are they comparable?

Our reply: Thank you for your question. Ecosystem respiration, soil temperature and moisture were measured in one 100 × 100 m plot and with three replicates under each land surface. Soil and vegetation were measured in all three 100 × 100 m plots. Each 100 × 100 m plot was in a distance of less than 50 m, which has the similar plant and terrain. We therefore believed they were comparable.

(9) L126, "were" logged . . .

Our reply: Thank you for your suggestion. We have changed "The Data logged automatically every 30 min" to "The data were logged automatically every 30 min" according to your suggestion.

(10) L129, soil hardness is not a very familiar concept. Explain it and what unit is used?

Our reply: Thank you for your suggestion. We have changed "soil hardness" to "soil compactness" according to your suggestion. We also added it unit both in result and Figure 5.

"Soil compactness was over 0.30 Pa in intact grassland patch and above pika tunnel, approximate 0.20 Pa for bald patches and less than 0.10 Pa for pika pile (Figure 5), respectively. "

[Figure]

Figure 5. Soil compactness under different surface types: (1) large bald patch (LP), (2) medium bald patch (MP), (3) small bald patch (SP), (4) intact grassland patch (IG), (5) above pika tunnel (PT) and (6) old pika pile (PP).

(11) L131, since the respiration measurement is the key of this study, more details are needed. How big is the chamber? Transparent of opaque? How many replicates? Only one gas analyzer was used? How many minutes did one measurement take? What is the frequency of the data? During which period (specific dates) were the measurements taken? Also, how the fluxes were calculated? How the air temperature inside of the chamber was measured?

Our reply: Thank you for your suggestion. We have added more information regarding ecosystem respiration measurement according to your suggestion (Line 133-155).

"Ecosystem respiration rates were measured using the LICOR-8150 Automated Soil $CO_2$ Flux System, which was an accessory for the LI-8100A with at most 8 individual chambers at one time. Ecosystem $CO_2$ emission was sampled and controlled by the LI-8100A Analyzer Control Unit. The air temperature inside of the chamber was measured using the internal thermistor of the chamber. The ecosystem $CO_2$ fluxes were calculated by the equation as follow.

$$Fc = \frac{10VP_0\left(1 - \dfrac{W_0}{1000}\right)}{RS(T_0 + 273.15)} \frac{\partial C'}{\partial t}$$

where $Fc$ is the soil $CO_2$ efflux rate ($\mu$mol m$^{-2}$ s$^{-1}$), $V$ is volume (cm$^3$), $P_0$ is the initial pressure (kPa), $W_0$ is the initial water vapor mole fraction (mmol mol$^{-1}$), $S$ is soil surface area (cm$^2$), $T_0$ is initial air temperature (°C), and $\partial C'/\partial t$ is the initial rate of change in water-corrected $CO_2$ mole fraction ($\mu$mol$^{-1}$ mol s$^{-1}$).

Six LICOR-8100-104 long-term opaque chambers (20cm in diameter LICOR, Inc., Lincoln, NE, USA) were used to measure alternately between three replicates for six land surface types. Therefore, 3 days at least were required to complete one rotation measurements of ecosystem respiration. To measure ecosystem respiration, eighteen polyvinyl chloride collars with a 20 cm inner diameter and a 12 cm height were inserted into the soil with 3-4 cm exposed to the air (Qin et al., 2013). All of the collars were installed at least 24 h before the first measurement to reduce disturbance-induced ecosystem $CO_2$ effluxes. Ecosystem respiration rates were measured every 7-10 days from June 16 to August 20 in 2016 depending on weather conditions. A round-the-clock measurement protocol was carried out and ecosystem respiration rates were measured every 30 minutes. Each measurement takes 1 minute and 45 seconds, including pre-purge 10 seconds, dead band 15 seconds, observation length 1 minute and post-purge 20 seconds."

(12) L138 change "determined" to "collected".

Our reply: Thank you for your suggestion. We have changed "determined" to "collected" according to your suggestion.

(13) L142 from each surface type?

Our reply: Thank you for your careful review. The sentence has changed to "Another five soil cores were sampled by cylindrical cutting ring (7 cm in diameter and 5.2 cm in depth) to determine soil bulk density from each land surface type." according to your suggestion.

(14) L149 how many replicates?

Our reply: Thank you for your careful review. Soil and vegetation samples were collected under six land surface types with three replicates in three 100 × 100 m plots. To eliminate the confusion, we have revised this part as follow (Line 171-176).

"There were a total of 108 aboveground and belowground vegetation samples (3 plots × 6 land surface types × 3 replicates) from the study area. Aboveground biomass was determined by clipping all above-ground living plants at ground level, drying (oven-dried at 65°C for 48 h) and weighing. Belowground biomass was sampled by collecting five soil columns, and each soil column was 5 cm in diameter and 40 cm in depth."

(15) L150 change "sampled" to "determined"

Our reply: Thank you for your careful review. We have changed "sampled" to "determined" according to your suggestion (Line 173).

(16) L152 each type?

Our reply: Thank you for your careful review. It means each soil columns. To eliminate the confusion, this sentence was changed to "There were a total of 108 aboveground and belowground vegetation samples (3 plots × 6 land surface types × 3 replicates) from the study area. Aboveground biomass was determined by clipping all above-ground living plants at ground level, drying (oven-dried at 65°C for 48 h) and weighing. Belowground biomass was sampled by collecting five soil columns, and each soil column was 5 cm in diameter and 40 cm in depth." (Line 171-176)

(17) L169, according to your figure, this seems like correlation analysis instead of regression.

Our reply: Thank you for your careful review. We have changed "regression analysis" to "correlation analysis" according to your suggestion.

(18) Figure 2, which year? Average Ta?

Our reply: Thank you for your careful review. All data in this manuscript were collected in 2016. Ta was daily average air temperature. To eliminate confusion, the title of Figure 2 has been changed to "Figure 2. Daily average air temperature and precipitation of the study site in 2016."

(19) Figure 3, monthly average?

Our reply: Thank you for your question. Both soil temperature and soil moisture were monthly average. To eliminate confusion, the title of Figure 3 has been changed to "Figure 3. Monthly average soil temperature and soil moisture under different surface types: (1) large bald patch (LP), (2) medium bald patch (MP), (3) small bald patch (SP), (4) intact grassland patch (IG), (5) above pika tunnel (PT) and (6) old pika pile (PP)."

(20) Figure 8, μmol instead of umol

Our reply: Thank you for your suggestion. We have replaced "umol" by "μmol" according to your suggestion.

(21) Figure 9, this is not a good way to present correlation results. First, specify what analysis in the caption. Second, the full correlation table looks redundant as it presents two copies of each pair of variables. Also, correlation coefficients and P value need to be included. Was the correlation done across the different surface types?

Our reply: Thank you for your suggestion. We have redrawn Figure 9 according to your suggestion. And now it contained both the correlation coefficients and P value in one figure. The correlation 
[revised manuscript text omitted]

[Figure]

**Figure 2.**

[Figure]

**Figure 3**.

[Figure]

**Figure 4.**

[Figure]

**Figure 5.**

[Figure]

**Figure 6.**

[Figure]

**Figure 7.**

[Figure]

**Figure 8.**

[Figure]

**Figure 9.**

[Figure]

---

## Author Comment (AC5) · 16 Oct 2018

Response to Anonymous Referee #1's comments:

General comments: The plateau pika (Ochotona curzoniae) is one of the main native soil faunas on the Qinghai-Tibet Plateau and plays a key role in the terrestrial ecosystem there. Previous studies have mainly focused on its active habits and the influence of population density on soil properties, plant communities, and so on. On contrast, the present study aims to study the effect of plateau pika disturbance and patchiness on ecosystem carbon emission at the plot scale (i.e. large bald patch, medium bald patch, small bald patch, intact grassland, above pika tunnel and pika pile). The results are critical for ecological restoration and environmental change on the Qinghai-Tibetan Plateau.

Our reply: We appreciate your positive comments. We have accepted all of the your suggestions and explained how we had revised the manuscript point by point.

Specific comments:

(1) Introduction: This section has not clarified clearly why we should study the effect of plateau pika disturbance and patchiness on ecosystem carbon emission at the plot scale, but not at other scales? What are the exact differences between this study and so many previous studies?

Our reply: Thank you for your careful review. We have revised the whole manuscript to eliminate the reader's confusion. In fact, we mainly focused on the effect of plateau pika disturbance and patchiness on ecosystem carbon emission of alpine meadow and we did not refer to any scales in this work. We have clarified clearly why we should study the effect of plateau pika disturbance and patchiness on alpine grassland at plot scale in our previous study (Yi et al., 2016).

Typically, most of the previous studies compared carbon fluxes under intact vegetation at plots with different number of pika burrows. However, ecosystem carbon emissions from the homogeneous land surface induced by pika piles and patchiness have yet to be quantified. These are the exact differences between this study and so many previous studies. We have revised this section as follow (Line 86-90).

"Previous studies have demonstrated that pikas disturbance and patchiness weaken the function of alpine meadow as a carbon sink (Liu et al., 13; Peng et al., 2015; Qin et al., 2018) and accelerated ecosystem carbon emission rate (Qin et al., 2015a). Nevertheless, most of these studies have mainly focused on ecosystem carbon emission rate under the homogeneous land surface rather than heterogeneous land surfaces."

Yi, S., Chen, J., Qin, Y., Xu, G.: The burying and grazing effects of plateau pika on alpine grassland are small: a pilot study in a semiarid basin on the Qinghai-Tibet Plateau, Biogeosciences, 13(22), 6273-6284, 2016.

(2) Materials and methods: Line 114-118: Is there any standard to distinguish the six representative underlying surfaces? Especially how to determine the threshold area for the division of large, medium and small bald patches (i.e. 9 $m^2$ and 1 $m^2$)?

Our reply: Thank you for your careful review. Six representative underlying surfaces were selected according to the previous work in our study site (Yi et al., 2016; Qin et al., 2018). They were distinguished easily in aerial photographs. Large bald patches had less vegetation cover and the smallest side was larger than 3 m. Medium patches also covered by less vegetation cover and the larger side was in a range of 1 to 3 m and small bald patches were characteristic by less vegetation cover and the larger side was less than 1 m. Intact grassland was characteristic by high vegetation cover and no large and medium bare land was found. Pika tunnel and pika pile usually co-exited. Pika tunnel is approximately 6 cm in diameter and pika pile is in the front of pika tunnel, 60 cm in diameter and less vegetation cover.

We calculated the threshold area of large, medium and small patches by aerial photograph. Each aerial photograph has 12 million pixels. At a height of 20 m, the resolution of each pixel is ~1 cm and each photograph covers ~26 m × 35 m of ground. Pixels in each aerial image were first classified into two groups, i.e. vegetated or bare patches (Yi, 2016). Then patches with different sizes were created using OpenCv Library. And finally, fractions of vegetation and bare patches (large, medium and small patches) were calculated. We revised this part as follow (Line 114-118).

"At early June 2016, three 100 m $\times$ 100 m plots were established as replicates. In each plot, six representative land surfaces were selected: (1) large bald patch with size larger than 9.0 m$^2$ (LP), (2) medium bald patch with size of 1.0-9.0 m$^2$ (MP), (3) small bald patch with size of less than 1.0 m$^2$ (SP), (4) intact grassland patch (IG), (5) above pika tunnel (PT), (6) old pika pile (PP) (Figure 1) (Yi et al., 2016; Qin et al., 2018)."

Yi, S.H., 2016. FragMAP: a tool for long-term and cooperative monitoring and analysis of small-scale habitat fragmentation using an unmanned aerial vehicle. Int. J. Remote Sens. 1-12. http://dx.doi.org/10.1080/01431161.2016.1253898.

Yi, S., Chen, J., Qin, Y., Xu, G.: The burying and grazing effects of plateau pika on alpine grassland are small: a pilot study in a semiarid basin on the Qinghai-Tibet Plateau, Biogeosciences, 13(22), 6273-6284, 2016.

Qin, Y., Yi, S., Ding, Y., Xu, G., Chen, J., Wang, Z.: Effects of small-scale patchiness of alpine grassland on ecosystem carbon and nitrogen accumulation and estimation in northeastern qinghai-tibetan plateau, Geoderma, 318, 52-63, 2018.

(3) Line 124-136: Were the soil temperature and moisture measured at all three 100 m × 100 m plots or only one 100 m × 100 m plots?

Our reply: Thank you for your question. Soil temperature and moisture were measured in one 100 m × 100 m plot where ecosystem respiration was measured. Both soil temperature and moisture were measured with three replicates under each underlying surface type. We revised this part to eliminate the confusion (Line 124-127).

"Soil temperature and moisture at 10 cm were measured in a 100 m × 100 m plot where ecosystem respiration was measured by using an auto-measurement system (Decagon Inc., USA) from early June to the late August. The system consisted of an EM50 logger and five 5TM sensors. The data were logged automatically every 30 min"

(4) Were the soil saturated hydraulic conductivity, soil hardness and ecosystem respiration rates measured for only one time or many times during the study periods? These key questions should be clarified.

Our reply: Thanks for your suggestion. Soil saturated hydraulic conductivity and soil hardness under each surface type were measured one time every month from June to August. Ecosystem respiration was measured every 7-10 days from June 16 to August 20 depending on weather conditions. We therefore revised this part as follow (Line 124-155).

"Soil temperature and moisture at 10 cm were measured in a 100 m × 100 m plot where ecosystem respiration was measured by using an auto-measurement system (Decagon Inc., USA) from early June to the late August. The system consisted of an EM50 logger and five 5TM sensors. The data were logged automatically every 30 min. Soil saturated hydraulic conductivity and compactness were measured once each month from June to August. Soil saturated hydraulic conductivity was measured by Dual Head infiltrometer (Decagon Inc., USA). The measurement process included 15 min soak time, 20 min hold time at low pressure head (5 cm) and high pressure head (15 cm) with 2 cycles. Each measurement takes 95 min altogether. Soil compactness was measured with TJSD-750 (Hangzhou Top Instrument co., LTD, Hangzhou, China) from the soil surface to 10 cm depth. Ecosystem respiration rates were measured using the LICOR-8150 Automated Soil $CO_2$ Flux System, which was an accessory for the LI-8100A with at most 8 individual chambers at one time. Ecosystem $CO_2$ emission was sampled and controlled by the LI-8100A Analyzer Control Unit. The air temperature inside of the chamber was measured using the internal thermistor of the chamber. The ecosystem $CO_2$ fluxes were calculated by the equation as follow.

$$Fc = \frac{10 V P_0 \left(1 - \dfrac{W_0}{1000}\right)}{RS(T_0 + 273.15)} \frac{\partial C'}{\partial t}$$

where $Fc$ is the soil $CO_2$ efflux rate (μmol m$^{-2}$ s$^{-1}$), $V$ is volume (cm$^3$), $P_0$ is the initial pressure (kPa), $W_0$ is the initial water vapor mole fraction (mmol mol$^{-1}$), $S$ is soil surface area (cm$^2$), $T_0$ is initial air temperature (°C), and $\partial C'/\partial t$ is the initial rate of change in water-corrected $CO_2$ mole fraction (μmol$^{-1}$ mol s$^{-1}$).

Six LICOR-8100-104 long-term opaque chambers (20cm in diameter LICOR, Inc., Lincoln, NE, USA) were used to measure alternately between three replicates for six land surface types. Therefore, 3 days at least were required to complete one rotation measurements of ecosystem respiration. To measure ecosystem respiration, eighteen polyvinyl chloride collars with a 20 cm inner diameter and a 12 cm height were inserted into the soil with 3-4 cm exposed to the air (Qin et al., 2013). All of the collars were installed at least 24 h before the first measurement to reduce disturbance-induced ecosystem $CO_2$ effluxes. Ecosystem respiration rates were measured every 7-10 days from June 16 to August 20 in 2016 depending on weather conditions. A round-the-clock measurement protocol was carried out and ecosystem respiration rates were measured every 30 minutes. Each measurement takes 1 minute and 45 seconds, including pre-purge 10 seconds, dead band 15 seconds, observation length 1 minute and post-purge 20 seconds."

(5) Line 138-141: How depth was the pika tunnel? Did this depth limit the collection of soil core to 40 cm?

Our reply: Thanks for your question. We investigated pika tunnel by digging soil pole and the depth of pika tunnel was about 40cm. Therefore, it wasn't difficulty to collect soil core at depth of 40cm. We have revised this part as follow (Line 157-164).

"Soil samples were collected during the periods of late July to early August 2016. In each surface type of each plot, five soil cores were collected using a stainless-steel auger (5 cm in diameter) at depths of 0-10, 10-20, 20-30 and 30-40 cm, and bulked as one composite sample for each depth in each quadrat. Another five soil cores were sampled by cylindrical cutting ring (7 cm in diameter and 5.2 cm in depth) to determine soil bulk density from each land surface type. Pika tunnel was approximate 6 cm in diameter and 40 cm in depth. Therefore, soil samples were available to collect at depth of 40cm. Totally, 512 soil samples were collected."

(6) Discussion: Line 216-217: "Nevertheless, the increased water infiltration was unable to increase soil moisture under pika pile." Why? The potential reasons should be discussed.

Our reply: Thanks for your suggestion. We discussed the reason why the increased water infiltration was unable to increase soil moisture under pika pile as follow (Line 263-271).

"Nevertheless, the increased water infiltration was unable to increase soil moisture under pika piles. For example, soil moisture under pika piles was approximate 5 % lower than intact grassland (Figure 4). Our result was discrepant with previous studies which reported old pika mound had the highest soil moisture during the summer (Ma et al., 2018) and moderate pika burrowing activities increased surface soil moisture (Li and Zhang, 2006). This difference may be contributed to the high pika density in alpine meadow (Guo et al, 2017). Moreover, pika piles were loose (Figure 6) with less vegetation cover (Figure 8), which was not beneficial for soil moisture storage."

(7) Line 227-229: The explanation for the low soil moisture under bald patches was not convincing, because the vegetation transpiration at intact grassland may be higher than the corresponding soil evaporation under bald patches at the same periods.

Our reply: Thanks for your comment. In fact, we have measured evaporation under different surfaces of the intact grassland, isolate grassland, large patches, medium patches and small patches since the early June 2016. It is difficult to measure evaporation from pika tunnle and pika pile due to their small sizes. Therefore, these data were not presented in this manuscript. We found that the evaporation under bald patches were higher than the intact grassland in our study sites through three years observation. We have revised this part as follow (Line 288-297).

"Our results showed that soil moisture under large and medium patches decreased 10 % than intact grassland (Figure 4). Previous studies had reported that the soil compaction of bald patches decreased the rate of water infiltration (Wuest et al., 2006; Wilson and Smith, 2015), which was similar with our results showed that bald patches had less saturated soil hydraulic conductivity (Figure 5). Low vegetation cover under bald patches was not beneficial for water retention and utilization, where most of soil water was mainly lost as a way of evaporation (Yi et al., 2014). We have measured evaporation of the intact grassland, isolate grassland, large patches, medium patches and small patches since the early June 2016. Three years results indicated that evaporation under bald patches were higher than the intact grassland (data were not shown here)."

(8) Line 230-233: More details about the reason for the different soil temperature patterns should be added.

Our reply: Thank you for your suggestion. We have added more detailed information about the difference of soil temperature between intact grassland and pika pile and bald patches. This part has been revised as follow (Line 301-309).

"Our results indicated that soil temperature under pika piles and bald patches was approximate 1 to 3 °C higher than intact grassland (Figure 4), which mainly resulted from the heterogeneity of surface albedo, surface soil water retention, heat conduction properties and radiation (Beringer et al., 2005; Pielke, 2005; Yi et al., 2013; You et al., 2017). It was suggested that pikas disturbance create a better soil temperature buffer for them to avoid the extreme cold in winter (Ma et al., 2018), whereas high soil temperature under bald patch was a disadvantage for the recovery of vegetation because patch surface had the smallest soil moisture content (Figure 4) and the largest daily range of soil temperature (Ma et al., 2018)."

(9) Line 234-235: What is the reason for the description of "high soil temperature under bald patch was a disadvantage for the recovery of vegetation"?

Our reply: Thank you for your question. Our study site belongs to semi-arid region, where water was one of dominant limit factors for vegetation growth. Patch surface had the smallest soil moisture content and the largest daily range of soil temperature, which was not beneficial for soil water retention. We have changed this part as follow (Line 305-309).

"It was suggested that pikas disturbance create a better soil temperature buffer for them to avoid the extreme cold in winter (Ma et al., 2018), whereas high soil temperature under bald patch was a disadvantage for the recovery of vegetation because patch surface had the smallest soil moisture content (Figure 4) and the largest daily range of soil temperature (Ma et al., 2018)."

Ma, Y.J., Wu, Y.N., Liu, W.L., Li, X.Y., Lin, H.S.: Microclimate response of soil to plateau pika's disturbance in the northeast qinghai-tibet plateau, European Journal of Soil Science, 69(2), 232-244, 2018.

Technical corrections:

(1) Line 33: Delete "under".

Our reply: Thank you for your suggestion. We have deleted "under" according to your suggestion.

(2) Line 88-90: This sentence is not exact, because lots of previous researches have studied the heterogeneous underground vegetation and belowground soil properties.

Our reply: Thank you for your suggestion. We totally agree with your comment that lots of previous researches have studied the heterogeneous underground vegetation and belowground soil properties. However, few studies have investigated the difference of ecosystem respiration under the heterogeneous underlying surface. Therefore, we have changed this sentence to "Nevertheless, most of these studies have mainly focused on ecosystem carbon emission rate under the homogeneous land surface rather than heterogeneous land surfaces." (Line 88-90)

(3) Line 188-189: This sentence has the same mean with the sentence in line 185-186.

Our reply: Thank you for your suggestion. We have deleted this sentence according to your suggestion.

(4) Line 197-198: According to the description in line 172, the growing season in the study is from May to September. Please add the data about ecosystem respiration in May and September.

Our reply: Thank you for your suggestion. Actually, our field observation started at the early June and finished at the late August in 2016. It's pity we can't add the data of ecosystem respiration in May and September.

(5) Line 214: Change "Figure 3" to "Figure 4".

Our reply: Thank you for your suggestion. We have changed "Figure 3" to "Figure 4" according to your suggestion.

(6) Line 311: Some references cited in the text were not listed in the "Reference" section.

Our reply: Thank you for your suggestion. The references have been checked carefully through manuscript according to your suggestion. And now all the references cited in the manuscript are also included in the "Reference" section.

(7) Line 518-520: Six small photos below the aerial photo are not clear. Moreover, add "MP" after "2".

Our reply: Thank you for your suggestion. We have redrawn Figure 1 according to your suggestion. We also add "MP" after "2". We believe that the photos are clear now.

[Figure]

Figure 1. An aerial photo of field observation of ecosystem respiration at six surface types: (1) Large bald patch (LP), (2) Medium bald patch (MP), (3) Small bald patch (SP), (4) Intact grassland patch (IG), (5) above pika tunnel (PT) and (6) old Pika pile (PP).

(8) Line 539: The regression analysis was used to analyze the relationships of ecosystem respiration with biotic and abiotic factors (line 168-169). However, the result in figure 9 was only the correlation coefficient between them.

Our reply: Thank you for your suggestion. We have redrawn Figure 9 according to your suggestion and now it contained both the correlation coefficients and P value in one figure. The title of Figure 9 was changed to "
[revised manuscript text omitted]

[Figure]

**Figure 2.**

[Figure]

**Figure 3**.

[Figure]

**Figure 4.**

[Figure]

**Figure 5.**

[Figure]

**Figure 6.**

[Figure]

**Figure 7.**

[Figure]

**Figure 8.**

[Figure]

**Figure 9.**

[Figure]

---

## Author Response (AR2)

**Reviewer 1**

This version improved and most questions raised by the initial review were answered. However, there are still some corrections to be made (as follows).

Our reply: We appreciate your positive comments. We have accepted all of the your suggestions and explained how we had revised the manuscript point by point.

Line 132-133: It showed that "For each surface type, nine 1 m × 1 m quadrats were set up, of which three was used for soil temperature and soil moisture measurement". However, "Soil temperature and moisture at 10 cm were measured in a 100 m × 100 m plot" in line 138. This means that three 1 m × 1 m quadrats for soil temperature and soil moisture measurement were set in the same 100 m × 100 m plot? Please clarify.

Our reply: Thanks for your careful review. We measured soil temperature and soil moisture in a 100 m × 100 m plot where ecosystem respiration was measured. To eliminate the confusion, we revised this part as follow (Line 132-143).

"For each surface type in each plot, six 1 m × 1 m quadrats were set up, of which three was used for soil saturated hydraulic conductivity measurement and three for soil compactness measurement, soil and vegetation sampling. We also set up another three 1 m × 1 m quadrats and three 2 m × 2 m quadrats in each surface type in a 100 m × 100 m plot for measuring soil temperature, soil moisture and ecosystem respiration."

"A meteorological tower was established in our observation station since 2008. Air temperature (°C) at 2.0m was measured by HMP45C (Vaisala, Helsinki, Finland), and precipitation was measured using an all-weather precipitation gauge (Geonor T-200B, Norway) (Wu et al., 2015). Soil temperature and moisture at 10 cm were measured by using an auto-measurement system (Decagon Inc., USA) from early June to the late August"

Line 153: What's the mean of "R"?

Our reply: Thanks for your careful review. $R$ is the ideal gas constant. We have revised this part as follow to eliminate confusion (Line 157-161).

$$Fc = \frac{10VP_0\left(1 - \frac{W_0}{1000}\right)}{RS(T_0 + 273.15)} \frac{\partial C'}{\partial t} \qquad (1)$$

where $Fc$ is the soil $CO_2$ efflux rate ($\mu$mol m$^{-2}$ s$^{-1}$), $V$ is volume (cm$^3$), $P_0$ is the initial pressure (kPa), $W_0$ is the initial water vapor mole fraction (mmol mol$^{-1}$), $R$ is the ideal gas constant, $S$ is soil surface area (cm$^2$), $T_0$ is initial air temperature (°C), and $\partial C'/\partial t$ is the initial rate of change in water-corrected $CO_2$ mole fraction ($\mu$mol$^{-1}$ s$^{-1}$).

Line 157: The unit is not correct.

Our reply: Thanks for your careful review. We have revised the unit of the initial rate of change in water-corrected $CO_2$ mole fraction to $\mu$mol$^{-1}$ s$^{-1}$ (Line 157-161).

Line 195-197: The number of these two equations should be (2) and (3), respectively.

Our reply: Thanks for your careful review. We have revised these two equations to 2 and 3 (Line 198-205).

The soil organic C and total N densities in different land surface were calculated using the equation (2) and (3):

$$SOC = \sum_{i=1}^{n} \rho * (1 - \sigma_{gravel}) * C_{SOC} * D_i \qquad (2)$$

$$TN = \sum_{i=1}^{n} \rho * (1 - \sigma_{gravel}) * C_{TN} * D_i \qquad (3)$$

where SOC is soil organic C density (kg m$^{-2}$), TN (kg m$^{-2}$) is soil total N density, $\rho$ is the soil bulk density (g cm$^{-3}$), $\sigma_{garvel}$ is the relative volume of gravel (% w/w), $C_{SOC}$ is soil organic C content (g kg$^{-1}$), $C_{TN}$ is soil total N content (g kg$^{-1}$) and Di is soil thickness (cm) at layer i, respectively; i=1, 2, 3 and 4.

Line 206: Which type of "correlation analysis" was used? Pearson or others.

Our reply: Thanks for your careful review. Pearson correlation analysis was used to analyze the relationships of ecosystem respiration with biotic and abiotic factors. Therefore, we revised this section as to "The relationships of ecosystem respiration with biotic and abiotic factors were analyzed by Pearson correlation analysis using R." (Line 209-211).

Line 213-215: The ecosystem respiration under intact grassland was lower than that above pika tunnel in August (Figure 2c). How can "…higher than other surface types both in July and August"?

Our reply: Thanks for your careful review. The average Re under above pika tunnel was missed in the previous revised manuscript and thus caused the confusion. We have added this data and revised this section as follow (Line 214-223).

"Pikas disturbance had significant effect on ecosystem respiration in June and July (Table 1, P<0.05), while the significant effect of patchiness on ecosystem respiration was found in July and August (Table 1, P<0.05). During the growing season, ecosystem respiration maximized in August and minimized in June (Figure 2). In June, ecosystem respiration under intact grassland, above pika tunnel, small patch and pika pile had no significant difference and the lowest ecosystem respiration was found under large and medium patches (Figure 2). Average ecosystem respiration under intact grassland was 4.01 μmol m$^{-2}$ s$^{-1}$ in July, which was 24.35 % to 137.39 % higher than other surface types (Figure 2). In August, average ecosystem respiration were 4.07 μmol m$^{-2}$ s$^{-1}$ and 4.85 μmol m$^{-2}$ s$^{-1}$ for intact grassland and above pika tunnel, 2.59-3.81 μmol m$^{-2}$ s$^{-1}$ for bald patches and 1.18 μmol m$^{-2}$ s$^{-1}$ for pika pile (Figure 2). "

Line 216: Change to "Insert Table 1, Figure 2 here".

Our reply: Thanks for your careful review. We have revised this as to "Insert Table 1, Figure 2 here" (Line 224).

Line 218-219: How the air temperature and rainfall were measured should be added in the "Field observation" section.

Our reply: Thanks for your careful review. We have added the measurement of air temperature and rainfall in the "Field observation" section according to your suggestion (Line 139-142).

"A meteorological tower was established in our observation station since 2008. Air temperature (°C) at 2.0m was measured by HMP45C (Vaisala, Helsinki, Finland), and precipitation was measured using an all-weather precipitation gauge (Geonor T-200B, Norway) (Wu et al., 2015)."

Line 226-229: There may be some mistakes of the data, because it was not consistent with the result in Figure 5. Please check.

Our reply: Thanks for your careful review. We have checked the data of soil saturated hydraulic conductivity and no mistake was found. Soil saturated hydraulic conductivity of intact grassland had no significant difference with small patch and above pika tunnel (P>0.05), soil saturated hydraulic conductivity under intact grassland, small patch and above pika tunnel were 2.13, 2.14 and 2.12 cm h$^{-1}$, respectively. Soil saturated hydraulic conductivity of intact grassland was approximate 40 % higher than medium and large patches and 17 % lower than pika pile. We have revised this section to eliminate the confusion (Line 233-236).

"Soil saturated hydraulic conductivity had no significant difference among different land surfaces (Table 2, P>0.05). However, soil saturated hydraulic conductivity under intact grassland was approximate 40 % higher than medium and large patches and 17 % lower than pika pile (Figure 5)."

Line 230: Change "Table 1" to "Table 2".

Our reply: Thanks for your careful review. We have changed "Table 1" to "Table 2" according to your suggestion (Line 237).

Line 241: Delete "Table 2".

Our reply: Thanks for your careful review. We have deleted "Table 2" according to your suggestion.

Line 260-261: Why "SOC and TN densities under pika pile decreased 13.35 % and 42.93 % than intact grassland" should be explained.

Our reply: Thanks for your careful review. We explained the reason of why "SOC and TN densities under pika pile declined as follow (Line 268-271).

"We also found that SOC and TN densities under pika pile decreased 13.35 % and 42.93 % than intact grassland. This was because pika burrowing activity transferred of deeper, nutrient-poor soil to the soil surface, improved soil aeration increased rate of organic carbon mineralization and soil erosion took away soil nutrition (Wei et al., 2006; Qin et al., 2015a; Chen et al., 2017)."

Line 323-325: More discussion about the effect of soil temperature on ecosystem respiration is necessary.

Our reply: Thanks for your careful review. We have revised this section according to your suggestion (Line 333-336).

"It was well known that rising of soil temperature under natural condition enhanced ecosystem respiration by stimulating decomposition of soil organic matter (Conant et al., 2008), increasing plant biomass (Yi et al., 2014) and activity of microbial enzymes (Bond-Lamberty and Thomson, 2010). However, obvious relationship between Re and soil temperature was not found in the present study (Figure 9), which suggested that other factors involved in controlling Re induced by pikas disturbance and patchiness."

Line 364: Add the page number.

Our reply: Thanks for your careful review. We have added the page number according to your suggestion (Line 401-402).

"Ahlström, A., Xia, J., Arneth, A., Luo, Y., Smith, B.: Importance of vegetation dynamics for future terrestrial carbon cycling, Environ. Res. Lett., 10(5), 1-11, 2015."

Line 564: Add the volume number.

Our reply: Thanks for your careful review. We have added the volume number according to your suggestion (Line 626-628).

"Yi, S.: Fragmap: a tool for long-term and cooperative monitoring and analysis of small-scale habitat fragmentation using an unmanned aerial vehicle, Int. J. Remote Sens., 38(8-10), 2686-2697, 2017."

Line 602-604: What's the mean of different lines and dot should be clarify.

Our reply: Thanks for your careful review. We have revised this part according to your suggestion. We have explained the mean of different lines and dot according to your suggestion (Line 670-675).

**Figure 2**. Ecosystem respiration of different surface types: (1) large bald patch (LP), (2) medium bald patch (MP), (3) small bald patch (SP), (4) intact grassland patch (IG), (5) above pika tunnel (PT) and (6) old pika pile (PP). The upper solid lines, the bottom solid lines, the bold solid horizontal line and the empty dot mean the maximum value, minimum value, median and abnormal value. Letters on the error bars indicate significant differences among different surface types at P < 0.05.

[Figure]

Line 627-628: You can use "**indicates significant differences at P < 0.01, * indicates significant differences at P < 0.05" to make it more clear.

Our reply: Thanks for your careful review. We have revised this part according to your suggestion (Line 693-701).

"Figure 9. The correlation coefficient charts between ecosystem respiration (Re) and biotic and abiotic factors for all six land surfaces. The diagonal line in the figure shows the distributions of the variables themselves. The red line means the frequency distribution of variables. The lower triangle (the left bottom of the diagonal) in the figure shows scatter plots of the two properties. The upper triangle (the upper right of the diagonal) in the figure indicates the correlation values of the two parameters; the asterisk indicates the degree of significance (*** indicates significant differences at P < 0.001, ** indicates significant differences at P < 0.01, * indicates significant differences at P < 0.05.). The bold bigger numbers mean the higher correlation."

Line 652: What the mean of the red line? I don't think the red line along the diagonal line is necessary.

Our reply: Thanks for your careful review. The red line means the frequency distribution of variables, which has the similar function with the histogram. However, it coexists with the histogram. If we delete the red line, the histogram would also disappear. We therefore used both the histogram and the red line and explain it in Figure 9 (Line 693-701).

"Figure 9. The correlation coefficient charts between ecosystem respiration (Re) and biotic and abiotic factors for all six land surfaces. The diagonal line in the figure shows the distributions of the variables themselves. The red line means the frequency distribution of variables. The lower triangle (the left bottom of the diagonal) in the figure shows scatter plots of the two properties. The upper triangle (the upper right of the diagonal) in the figure indicates the correlation values of the two parameters; the asterisk indicates the degree of significance (*** indicates significant differences at $P < 0.001$, ** indicates significant differences at $P < 0.01$, * indicates significant differences at $P < 0.05$.). The bold bigger numbers mean the higher correlation."

**Reviewer 3**

Plateau pika disturbance can be a biotic factor that contributes to the different patchiness, such as large patchiness, medium patchiness.... So, the topic of this manuscript should focus on comparing ecosystem carbon emission among different patchiness, and disscuss the all possible factors both biotic and abitic that can cause patchiness such plateau pika, zokor, marmot, livestock, permofrost.....

Our reply: We appreciate your constructive comments. We have explained how we had revised the manuscript point by point.

Firstly, patchiness in alpine meadow are always induced by multiple factors, such as, grazing, plateau pika disturbance, zokor disturbance, marmot disturbance, permafrost degradation, etc, that is, there is relationship between plateau pika disturbance and patchiness. Plateau pika disturbance also can contribute to the large, medium and small bald patches in your manuscript, because plateau pika dose not has the digging activities, but also has burying activities. However, in your manuscript, it seems that plateau pikas disturbance and patchiness were two different factors. it is confusing. How did authors distinguish the effects of these two factors? As a result the treatment is potentially confounded with other conditions (permafrost, grazing, other small mammals). This is a fatal problem that authors confuse the plateau pika disturbance and patchiness.

Our reply: We appreciate your constructive comments. As indicated by your comments, the patchiness of alpine grassland originates from multiple factors. We agree that "there is relationship between plateau pika disturbance and patchiness", however, it is not our aim to investigate whether bald patchiness originates from plateau pika or is affected by other conditions (permafrost, grazing, other small mammals). To make our point clear, we modified our aims at end of Introduction (Line 90-94). We also discussed the origination of patchiness in Discussion (Line 354-378). Thus, the specific aims of this study were to (1) investigate the spatial heterogeneity of Re among different surface types (plateau pika pile, different sizes of bald patches and vegetation) of alpine grassland; (2) illuminate the potential regulating mechanism of pikas disturbance and patchiness to ecosystem respiration (Re) in an alpine meadow grassland in the northeastern part of Qinghai-Tibetan Plateau (QTP). We acknowledge the compounding effects of plateau pika (and also other factors, e.g. permafrost degradation, grazing, etc.) on patchiness, but it is not our aim to investigate the origination of patchiness. In the following part, when we mention pika effects, we mean the direct effects of piles and tunnels from pika excavating other than the bald batches originate from plateau pika or no.

"**Effect of pikas disturbance on patchiness**

Natural vegetation patches, bald patches with different sizes and pikas piles coexisted on the alpine meadow (Figure 1), which supported that alpine grassland had also experienced fragmentation (Qin et al., 2018). Several proposed mechanisms may be accounted for the formation and development of patchiness in alpine grassland. As one of dominant form of land utilization, alpine grasslands are widely used for grazing. Previous studies suggested that overgrazing destroyed the original vegetation and led to decrease in the coverage and looseness of soil (Dong et al., 2013), which was prone to form bald patch due to soil erosion (Fécan et al., 1998; Zhang and Dong, 2014). Other than livestock, alpine grassland is also habitats for many small mammals such as plateau pika, zokor (*Eospalax fontanierii*), marmot (*Marmota himalayana*) and fox (*Vulpes ferrilata*). Pikas were considered to create a patchy matrix by changing soil properties (Chen et al., 2017), digging tunnels and burying activities (Dong et al., 2013). On one hand, pikas bury vegetation by fresh excavated soil, then small bare soil patches are formed and further large soil patches are then formed by linking small bare soil patches by wind and/or water (Wei et al., 2007; Ma et al., 2018). On the other hand, pikas dig tunnel underground. Although pikas make burrows are the primary homes to a wide variety of small birds and lizards (Smith and Foggin, 1999), the collapse of pika tunnels results in the emergence of bald soil patches (Zhou et al., 2003; Cao et al., 2010). Moreover, alpine grassland is underlain by extensive permafrost (Chen and Wu, 2007). The repeated freeze and thaw cause the crack of the sod around the barren area (Yang et al. 2003) and create precondition for forming bald patch. However, to date, there are no direct evidences to demonstrate the potential mechanism for forming and developing of patchiness for alpine grassland on the QTP. It is, therefore, critical to perform long-term repeated monitoring studies to determine whether bald patches are developed from pika piles or burrow tunnels and what the major factors affecting bald patch expansion are (Yi et al., 2016)."

Secondly, the size of plot in the manuscript is 10000 m$^2$ (100 m * 100 m). however, pikas are social mammals that live in family group, and the average home range is about 1,262.5 m$^2$ to 2,308 m$^2$, so, any other mammals in the plots? This is, pile could be contributed by other mammals, such as marmot, zokor…

Our reply: Thank you for your careful review. There are no other mammals, e.g. marmot and zokor in our study plots. All of the piles in each plot were created by plateau pikas. To eliminate the confusion, we added the detailed information in the Field observation (Line 113-127).

"At early June 2016, three 100 m $\times$ 100 m plots were established as replicates. Each 100 $\times$ 100 m plot was in a distance of less than 50 m, which has the similar plant and terrain. In each plot, six representative land surfaces were selected: (1) large bald patch with size larger than 9.0 m$^2$ (LP), (2) medium bald patch with size of 1.0-9.0 m$^2$ (MP), (3) small bald patch with size of less than 1.0 m$^2$ (SP), (4) intact grassland patch (IG), (5) above pika tunnel (PT), (6) old pika pile (PP) (Figure 1) (Yi et al., 2016; Qin et al., 2018). There are no other mammals, e.g. marmot, zokor in our study plots. All of the piles in each plot were created by plateau pikas. They were distinguished easily in aerial photographs. Large bald patches had less vegetation cover and the smallest side was larger than 3 m. Medium patches also covered by less vegetation cover and the largest side was in a range of 1 to 3 m and small bald patches were characterized by less vegetation cover and the largest side was less than 1 m. Intact grassland was characterized by high vegetation cover and no large and medium bare land was found. Pika tunnel and pika pile usually co-existed. Pika tunnel is approximately 6 cm in diameter and pika pile is in the front of pika tunnel, 60 cm in diameter and less vegetation cover."

As for statistics

1. One-way analysis of variance (ANOVA) and a multi-comparison of a least significant difference (LSD) test were used to determine differences at the p=0.05 level, however, from the manuscript, pika disturbance and patchiness are two independent factors, if so, authors should use two-way anova.

Our reply: Thank you for your suggestion. We have reanalyzed the data using two-way analysis of variance (ANOVA). The results were showed in table 1 and 2.

**Table 1.** Two-way ANOVA results of the effect of patches fragmentation and pikas disturbance on soil temperature, soil moisture and ecosystem respiration.

|  |  | Soil temperature | | | Soil moisture | | | Ecosystem respiration | | |
|---|---|---|---|---|---|---|---|---|---|---|
|  |  | Jun | Jul | Aug | Jun | Jul | Aug | Jun | Jul | Aug |
| Patchiness | $F$ | 10.44 | 20.63 | 3.51 | 218.23 | 205.44 | 62.56 | 7.03 | 18.98 | 2.71 |
|  | $P$ | <0.001 | <0.001 | 0.03 | <0.001 | <0.001 | <0.001 | 0.002 | <0.001 | 0.12 |
| Pikas disturbance | $F$ | 16.85 | 20.14 | 3.68 | 4.80 | 12.97 | 3.21 | 0.4 | 4.93 | 11.58 |
|  | $P$ | <0.001 | <0.001 | 0.03 | 0.012 | <0.001 | 0.05 | 0.68 | 0.023 | 0.009 |

**Table 2.** Two-way ANOVA results of the effect of patches fragmentation and pikas disturbance on soil compactness, aboveground biomass, belowground biomass, soil hydraulic conductivity, SOC and TN density.

|  |  | Soil compactness | Aboveground biomass | Belowground biomass | Saturated hydraulic conductivity | SOC density | TN density |
|---|---|---|---|---|---|---|---|
| Patchiness | $F$ | 28.10 | 12.15 | 7.24 | 0.75 | 4.49 | 10.78 |
|  | $P$ | <0.001 | 0.002 | 0.023 | 0.54 | 0.04 | 0.003 |
| Pikas disturbance | $F$ | 55.86 | 8.77 | 11.98 | 0.42 | 372.10 | 69.49 |
|  | $P$ | <0.001 | 0.017 | 0.002 | 0.67 | <0.001 | <0.001 |

Results: the results are rambling, a summary in each section was lacked.

line 232, "Both pikas disturbance and patchiness significantly affected soil compactness, SOC density, TN density and vegetation biomass", both? They are independent? it is difficult to find "both" are significant in the table 2.

Our reply: Thank you for your careful review. The significant difference of soil compactness, SOC density, TN density and vegetation biomass under different underlying surfaces were reanalyzed by using two-way analysis of variance (ANOVA). The results were showed in table 1 and 2.

**Table 1.** Two-way ANOVA results of the effect of patches fragmentation and pikas disturbance on soil temperature, soil moisture and ecosystem respiration.

|  |  | Soil temperature | | | Soil moisture | | | Ecosystem respiration | | |
|---|---|---|---|---|---|---|---|---|---|---|
|  |  | Jun | Jul | Aug | Jun | Jul | Aug | Jun | Jul | Aug |
| Patchiness | $F$ | 10.44 | 20.63 | 3.51 | 218.23 | 205.44 | 62.56 | 7.03 | 18.98 | 2.71 |
|  | $P$ | <0.001 | <0.001 | 0.03 | <0.001 | <0.001 | <0.001 | 0.002 | <0.001 | 0.12 |
| Pikas disturbance | $F$ | 16.85 | 20.14 | 3.68 | 4.80 | 12.97 | 3.21 | 0.4 | 4.93 | 11.58 |
|  | $P$ | <0.001 | <0.001 | 0.03 | 0.012 | <0.001 | 0.05 | 0.68 | 0.023 | 0.009 |

**Table 2.** Two-way ANOVA results of the effect of patches fragmentation and pikas disturbance on soil compactness, aboveground biomass, belowground biomass, soil hydraulic conductivity, SOC and TN density.

|  |  | Soil compactness | Aboveground biomass | Belowground biomass | Saturated hydraulic conductivity | SOC density | TN density |
|---|---|---|---|---|---|---|---|
| Patchiness | $F$ | 28.10 | 12.15 | 7.24 | 0.75 | 4.49 | 10.78 |
|  | $P$ | <0.001 | 0.002 | 0.023 | 0.54 | 0.04 | 0.003 |
| Pikas disturbance | $F$ | 55.86 | 8.77 | 11.98 | 0.42 | 372.10 | 69.49 |
|  | $P$ | <0.001 | 0.017 | 0.002 | 0.67 | <0.001 | <0.001 |

Discussion:

The discussion is very detailed and specific and repeats the results. Patchiness can be caused by several factors (other small mammals, large herbivore, permafrost degradation…etc), however, the authors consider pika disturbance as single factor. In fact, plateau pika disturbance can contribute to cause any patchiness, such as large patchiness, medium patchiness, small patchiness.

Our reply: Thank you for your suggestion. We completely agreed with that plateau pika disturbance may contribute to the large, medium and small bald patches. We therefore added one section in discussion to explain the potential contribution of pikas disturbance and other factors to patchiness (Line 354-378).

"**Effect of pikas disturbance on patchiness**

Natural vegetation patches, bald patches with different sizes and pikas piles coexisted on the alpine meadow (Figure 1), which supported that alpine grassland had also experienced fragmentation (Qin et al., 2018). Several proposed mechanisms may be accounted for the formation and development of patchiness in alpine grassland. As one of dominant form of land utilization, alpine grasslands are widely used for grazing. Previous studies suggested that overgrazing destroyed the original vegetation and led to decrease in the coverage and looseness of soil (Dong et al., 2013), which was prone to form bald patch due to soil erosion (Fécan et al., 1998; Zhang and Dong, 2014). Other than livestock, alpine grassland is also habitats for many small mammals such as plateau pika, zokor (*Eospalax fontanierii*), marmot (*Marmota himalayana*) and fox (*Vulpes ferrilata*). Pikas were considered to create a patchy matrix by changing soil properties (Chen et al., 2017), digging tunnels and burying activities (Dong et al., 2013). On one hand, pikas bury vegetation by fresh excavated soil, then small bare soil patches are formed and further large soil patches are then formed by linking small bare soil patches by wind and/or water (Wei et al., 2007; Ma et al., 2018). On the other hand, pikas dig tunnel underground. Although pikas make burrows are the primary homes to a wide variety of small birds and lizards (Smith and Foggin, 1999), the collapse of pika tunnels results in the emergence of bald soil patches (Zhou et al., 2003; Cao et al., 2010). Moreover, alpine grassland is underlain by extensive permafrost (Chen and Wu, 2007). The repeated freeze and thaw cause the crack of the sod around the barren area (Yang et al. 2003) and create precondition for forming bald patch. However, to date, there are no direct evidences to demonstrate the potential mechanism for forming and developing of patchiness for alpine grassland on the QTP. It is, therefore, critical to perform long-term repeated monitoring studies to determine whether bald patches are developed from pika piles or burrow tunnels and what the major factors affecting bald patch expansion are (Yi et al., 2016)."

Details:

table 1 and table 2 "ANOVA results of the effect of patches fragmentation and small mammal activities….." Small mammal, just plateau pika, or any other small mammals?

Our reply: Thank you for your suggestion. Small mammals only mean plateau pika in our study area. Thus, we have changed "small mammal activities" to "pikas disturbance" both in table 1 and 2 (Line 660-665).

"Table 1. Two-way ANOVA results of the effect of patches fragmentation and pikas disturbance on soil temperature, soil moisture and ecosystem respiration."

"Table 2. Two-way ANOVA results of the effect of patches fragmentation and pikas disturbance on soil compactness, aboveground biomass, belowground biomass, soil hydraulic conductivity, SOC and TN density."

To eliminate the confusion, we added the detailed information in the Field observation (Line 113-127).

"At early June 2016, three 100 m $\times$ 100 m plots were established as replicates. Each 100 $\times$ 100 m plot was in a distance of less than 50 m, which has the similar plant and terrain. In each plot, six representative land surfaces were selected: (1) large bald patch with size larger than 9.0 m$^2$ (LP), (2) medium bald patch with size of 1.0-9.0 m$^2$ (MP), (3) small bald patch with size of less than 1.0 m$^2$ (SP), (4) intact grassland patch (IG), (5) above pika tunnel (PT), (6) old pika pile (PP) (Figure 1) (Yi et al., 2016; Qin et al., 2018). There are no other mammals, e.g. marmot, zokor in our study plots. All of the piles in each plot were created by plateau pikas. They were distinguished easily in aerial photographs. Large bald patches had less vegetation cover and the smallest side was larger than 3 m. Medium patches also covered by less vegetation cover and the largest side was in a range of 1 to 3 m and small bald patches were characterized by less vegetation cover and the largest side was less than 1 m. Intact grassland was characterized by high vegetation cover and no large and medium bare land was found. Pika tunnel and pika pile usually co-existed. Pika tunnel is approximately 6 cm in diameter and pika pile is in the front of pika tunnel, 60 cm in diameter and less vegetation cover."

Fig. 2. a multi-comparison has been done, so what's the difference among the different surface types, different letters showing the differences among the different surface types were lacked.

Our reply: Thank you for your suggestion. We have added different letters above bar to show the differences of Re among the different surface types (Line 670-675).

[revised manuscript text omitted]

---

## Author Response (AR3)

Response to Anonymous Referee #1's comments:

Most questions raised by previous reviews have been answered. However, there are still some corrections to be made (as follows), which should be revised before this paper could be accepted.

Our reply: We appreciate your positive comments. We have accepted all of the your suggestions and explained how we had revised the manuscript point by point.

Line 91-92: The land surface type "pika tunnel" was lost.

Our reply: Thanks for your careful review. We have added "pika tunnel" according to your suggestion (Line 91-95).

"(1) investigate the spatial heterogeneity of Re among different surface types (plateau pika pile, above pika tunnel, different sizes of bald patches and vegetation) of alpine grassland; (2) illuminate the potential regulating mechanism of pikas disturbance and patchiness to ecosystem respiration (Re) in an alpine meadow grassland in the northeastern part of Qinghai-Tibetan Plateau (QTP)."

Line 159: What is the value and unit of R (ideal gas constant)?

Our reply: Thanks for your careful review. The unit of R (ideal gas constant) had been added according to your suggestion (Line 159-162).

"where $Fc$ is the soil $CO_2$ efflux rate ($\mu mol\ m^{-2}\ s^{-1}$), $V$ is volume ($cm^3$), $P_0$ is the initial pressure (kPa), $W_0$ is the initial water vapor mole fraction ($mmol\ mol^{-1}$), $R$ is the ideal gas constant ($J\ K^{-1} mol^{-1}$), $S$ is soil surface area ($cm^2$), $T_0$ is initial air temperature (°C), and $\partial C'/\partial t$ is the initial rate of change in water-corrected $CO_2$ mole fraction ($\mu mol^{-1}\ s^{-1}$). "

Line 214-216: The description was inconsistent with Table 1.

Our reply: Thanks for your careful review. We have revised this part as follow (Line 215-217).

"Ecosystem respiration showed significant difference among varied land surface types during the growing season (Table 1, $P < 0.001$). Except for the pika pile, ecosystem respiration maximized in August and minimized in June (Figure 2)."

Line 216-217: I don't think the description is correct. For example, the Re of PP in June was higher than that in August.

Our reply: Thanks for your careful review. We completely agree with you that Re of PP in

June was higher than that in August. To eliminate the confusion, we have revised this part as follow (Line 215-217).

"Ecosystem respiration showed significant difference among varied land surface types during the growing season (Table 1, P<0.001). Except for the pika pile, ecosystem respiration maximized in August and minimized in June (Figure 2)."

Line 234-236: Please add the value. I think there may be some mistakes, because the description was inconsistent with Figure 5.

Our reply: Thanks for your careful review. We have added the value of soil saturated hydraulic conductivity under different surface types according to your suggestion (Line 234-241).

"Soil saturated hydraulic conductivity also showed significant variation under different land surface types (P=0.027, Table 2). For example, soil saturated hydraulic conductivity under large bald patch, medium bald patch, small bald patch, intact grassland patch, above pika tunnel and old pika pile were 1.54, 1.53, 2.14, 2.13, 2.12 and 2.58 cm h$^{-1}$, respectively (Figure 5). Soil saturated hydraulic conductivity under intact grassland patch was approximate 40 % higher than medium and large patches and 17 % lower than pika pile, while there was no significant difference among intact grassland patch, small patch and above pika tunnel (P>0.05)."

Line 256-378: This study was conducted only in growing season (June-August), however, some conclusions of cited references in the "Discussion" section were done all the year round. This may lead to lots of inconsistency, which should be clarified.

Our reply: Thanks for your careful review. We completely agree with you that some of conclusions of cited references in the "Discussion" section were done all the year round and we get them in growing season. Due to harsh environment on the QTP, it is difficult to conduct long-term field observation of ecosystem respiration all the year round. However, the main conclusions of cited references about the effect of patchiness and plateau pika on ecosystem respiration (Liu et al., 2013; Peng et al., 2015; Qin et al., 2015a), soil nutrition (Li and Zhang, 2006; Chen et al., 2017; Qin et al., 2018), soil temperature and soil moisture (Ma et al., 2018), vegetation biomass (Yi et al., 2016), soil saturated hydraulic conductivity (Wilson and Smith, 2015) at alpine grasslands in this manuscript were all conducted in the growing season. We therefore believe our conclusions are comparable with the previous studies.

Chen, J., Yi, S., Qin, Y.: The contribution of plateau pika disturbance and erosion on patchy alpine grassland soil on the Qinghai-Tibetan Plateau: Implications for grassland restoration, Geoderma, 297, 1-9, 2017.

Li, W. and Zhang, Y.: Impacts of plateau pikas on soil organic matter and moisture content in alpine meadow, Acta. Theriol. Sin., 26(4), 331-337, 2006.

Liu, Y.S., Fan, J.W., Harris, W., Shao, Q.Q., Zhou, Y.C., Wang, N., Li, Y.Z.: Effects of plateau pika (Ochotona curzoniae) on net ecosystem carbon exchange of grass-land in the Three Rivers Headwaters region, Qinghai-Tibet, China, Plant. Soil., 366,491-504, 2013.

Ma, Y.J., Wu, Y.N., Liu, W.L., Li, X.Y., Lin, H.S.: Microclimate response of soil to plateau pika's disturbance in the northeast qinghai-tibet plateau, European Journal of Soil Science, 69(2), 232-244, 2018.

Peng, F., Quangang, Y., Xue, X., 111, J., Wang, T.: Effects of rodent-induced land degradation on ecosystem carbon fluxes in alpine meadow in the Qinghai-Tibet Plateau, China, Solid. Earth., 6, 303-310, 2015.

Qin, Y., Chen, J.J., Yi, S.H.: Plateau pikas burrowing activity accelerates ecosystem carbon emission from alpine grassland on the Qinghai-Tibetan Plateau, Ecol. Eng., 84, 287-291, 2015a.

Qin, Y., Yi, S., Ding, Y., Xu, G., Chen, J., Wang, Z.: Effects of small-scale patchiness of alpine grassland on ecosystem carbon and nitrogen accumulation and estimation in northeastern qinghai-tibetan plateau, Geoderma, 318, 52-63, 2018.

Wilson, M.C. and Smith, A.T.: The pika and the watershed: The impact of small mammal poisoning on the ecohydrology of the Qinghai-Tibetan Plateau, Ambio, 44(1), 16-22, 2015.

Yi, S., Chen, J., Qin, Y., Xu, G.: The burying and grazing effects of plateau pika on alpine grassland are small: a pilot study in a semiarid basin on the Qinghai-Tibet Plateau, Biogeosciences, 13(22), 6273-6284, 2016.

Line 434: Change "qinghai-tibet plateau" to "Qinghai-Tibet Plateau".

Our reply: Thanks for your careful review. We have changed "qinghai-tibet plateau" to "Qinghai-Tibet Plateau" according to your suggestion (Line 437-438).

Cheng, G., Wu, T.: Responses of permafrost to climate change and their environmental significance, Qinghai-tibet plateau, J. Geophys. Res., 112(F2), 1-10, 2007.

Line 486: Add all author name instead of "et al".

Our reply: Thanks for your careful review. We have added all author names in this reference (Line 490-497).

"Janssens, I. A., Lankreijer, H., Matteucci, G., Kowalski, A. S., Buchmann, N., Epron, D., Pilegaard, K., Kutsch, W., Longdoz, B., Grünwald, T., Montagnani, L., Dore, S., Rebmann, C., Moors, E.J., Grelle, A., Rannik, Morgenstern, K., Oltchev, S., Clement, R., Gudmundsson, J., Minerbi, S., Berbigier, P., Ibrom, A., Moncrieff, J., Aubinet, M., Bernhofer, C., Jensen, N.O., Vesala, T., Granier, A., Schulze, E.D., Lindroth, A., Dolman, A.J., Jarvis, P.G., Ceulemans, R., Valentini, R.: Productivity overshadows temperature in determining soil and ecosystem respiration across european forests, Global. Change. Biol., 7(3), 269-278, 2001."

Line 677: Add the depth of soil temperature and soil moisture.

Our reply: Thanks for your careful review. We have added the depth of soil temperature and soil moisture in Figure 4 (Line 685-687).

"Figure 4. Monthly average soil temperature and soil moisture at 10 cm depth under different surface types: (1) large bald patch (LP), (2) medium bald patch (MP), (3) small bald patch (SP), (4) intact grassland patch (IG), (5) above pika tunnel (PT) and (6) old pika pile (PP)."

Response to Anonymous Referee #2's comments:

The authors addressed most of my concerns.

Our reply: We appreciate your positive comments.

Response to Anonymous Referee #3's comments:

The quality of manuscript has been improved greatly in the revision. However, in the response, I find that you mentioned that the specific aims of this study were to (1) investigate the spatial heterogeneity of Re among different surface types (plateau pika pile, different sizes of bald patches and vegetation) of alpine grassland; ......this is ok, you say that plateau pika pile is just one of the surface types, so your question is one factor: surface types. But in data analysis you consider plateau pika disturbance as an independent factor to do TWO-WAY ANOVA. Consequently, the statistics is wrong.

Our reply: We appreciate your constructive comments. We completely agree with you that we have made wrong statistics by using TWO-WAY ANOVA. Actually, we used One-way analysis of variance (ANOVA) to determine the differences of variables among six surface types in the first revision. However, we misunderstood your comments in second revision and improperly used TWO-WAY ANOVA. Therefore, we reanalyzed the data by using ANOVA and revised the related section of "Statistical analysis", "Results" and "Table list".

**Statistical analysis** (Line 208-210)

"One-way analysis of variance (ANOVA) and a multi-comparison of a least significant difference (LSD) test were used to determine differences at the p=0.05 level."

**Results**

**Ecosystem respiration** (Line 215-217)

"Ecosystem respiration showed significant difference among varied land surface types during the growing season (Table 1, P<0.001). Except for the pika pile, ecosystem respiration maximized in August and minimized in June (Figure 2). "

**Microclimate and soil hydrothermal characteristics** (Line 234-241)

"Soil saturated hydraulic conductivity also showed significant variation under different land surface types (P=0.027, Table 2). For example, soil saturated hydraulic conductivity under large bald patch, medium bald patch, small bald patch, intact grassland patch, above pika tunnel and old pika pile were 1.54, 1.53, 2.14, 2.13, 2.12 and 2.58 cm h$^{-1}$, respectively (Figure 5). Soil saturated hydraulic conductivity under intact grassland patch was approximate 40 % higher than medium and large patches and 17 % lower than pika pile, while it was no significant difference among intact grassland patch, small patch and above pika tunnel (P>0.05)."

**Soil and vegetation properties** (Line 244-245)

"Soil and vegetation properties showed significant variation under different land surface types (Table 2) (P<0.001)."

**Table list** (Line 669-672)

Table 1. ANOVA results of soil temperature, soil moisture and ecosystem respiration under different land surface types.

| | Soil temperature | | | Soil moisture | | | Ecosystem respiration | | |
|---|---|---|---|---|---|---|---|---|---|
| | June | July | August | June | July | August | June | July | August |
| F | 8.614 | 10.955 | 1.806 | 387.472 | 210.878 | 97.060 | 5.270 | 10.447 | 8.855 |
| P | <0.001 | <0.001 | 0.106 | <0.001 | <0.001 | <0.001 | 0.001 | <0.001 | <0.001 |

Table 2. ANOVA results of soil compactness, aboveground biomass, belowground biomass, soil hydraulic conductivity, SOC and TN density under different land surface types.

| | Soil compactness | Aboveground biomass | Belowground biomass | Saturated hydraulic conductivity | SOC density | TN density |
|---|---|---|---|---|---|---|
| F | 81.506 | 6.193 | 12.925 | 2.752 | 145.942 | 50.567 |
| P | <0.001 | 0.002 | <0.001 | 0.027 | <0.001 | <0.001 |

By the way, if you consider patchiness and pika disturbance as two different independent factors, your table 1 and table 2 were incomplete, how about interaction of two factors.

Our reply: We appreciate your constructive comments. The treatments in this study do not meet the criteria for analyzing the interaction of patchiness and pika disturbance using TWO-WAY ANOVA. To eliminate the confusion, we have reanalyzed the data by ANOVA and revised the related part in the manuscript.

In conclusion, experimental design and data statistics at all are mismatching.

Our reply: We appreciate your constructive comments. The data were reanalyzed by ANOVA and the related sections were also revised. We believe experimental design and data statistics are consistent now.

[revised manuscript text omitted]